# Catching the Big Fish in Big Data: A Meta-Analysis of Zebrafish Kidney scRNA-Seq Datasets Highlights Conserved Molecular Profiles of Macrophages and Neutrophils in Vertebrates

**DOI:** 10.3390/biology13100773

**Published:** 2024-09-27

**Authors:** Aleksandr V. Bobrovskikh, Ulyana S. Zubairova, Ludmila G. Naumenko, Alexey V. Doroshkov

**Affiliations:** 1Department of Physics, Novosibirsk State University, 630090 Novosibirsk, Russia; l.naumenko@g.nsu.ru; 2The Federal Research Center Institute of Cytology and Genetics, Siberian Branch of the Russian Academy of Sciences, 630090 Novosibirsk, Russia; ulyanochka@bionet.nsc.ru (U.S.Z.); ad@bionet.nsc.ru (A.V.D.); 3Department of Information Technologies, Novosibirsk State University, 630090 Novosibirsk, Russia; 4Department of Genomics and Bioinformatics, Institute of Fundamental Biology and Biotechnology, Siberian Federal University, 660036 Krasnoyarsk, Russia

**Keywords:** innate immune system, transcriptomic meta-analysis, single-cell RNA sequencing, gene network, transcription factors

## Abstract

**Simple Summary:**

This paper presents a meta-analysis of currently available single-cell RNA sequencing (scRNAseq) datasets from zebrafish kidneys. Our work aims to identify key marker genes for various immune cell types and transcription factors (TFs) involved in fish myelopoiesis. Our results validate previous studies and expand them; newly discovered markers and TFs have experimental confirmation of their roles in complementary cell types of mammals and fish species. Reconstructed gene networks of macrophage- and neutrophil-specific genes revealed the potential interactions of identified TFs and marker genes. Obtained results could be taken into account during the design of immunogenetics experiments (e.g., creating zebrafish lines with TF knockouts) and serve as a basis for novel applications for zebrafish. The revealed cell-specific markers could help in the accurate determination of fish cell types in the future. Our methodology of data integration showed its reliability and could be further adapted to meta-analyses of various scRNAseq datasets.

**Abstract:**

The innate immune system (IIS) is an ancient and essential defense mechanism that protects animals against a wide range of pathogens and diseases. Although extensively studied in mammals, our understanding of the IIS in other taxa remains limited. The zebrafish (*Danio rerio*) serves as a promising model organism for investigating IIS-related processes, yet the immunogenetics of fish are not fully elucidated. To address this gap, we conducted a meta-analysis of single-cell RNA sequencing (scRNA-seq) datasets from zebrafish kidney marrow, encompassing approximately 250,000 immune cells. Our analysis confirms the presence of key genetic pathways in zebrafish innate immune cells that are similar to those identified in mammals. Zebrafish macrophages specifically express genes encoding cathepsins, major histocompatibility complex class II proteins, integral membrane proteins, and the V-ATPase complex and demonstrate the enrichment of oxidative phosphorylation ferroptosis processes. Neutrophils are characterized by the significant expression of genes encoding actins, cytoskeleton organizing proteins, the Arp2/3 complex, and glycolysis enzymes and have demonstrated their involvement in GnRH and CLR signaling pathways, adherents, and tight junctions. Both macrophages and neutrophils highly express genes of NOD-like receptors, phagosomes, and lysosome pathways and genes involved in apoptosis. Our findings reinforce the idea about the existence of a wide spectrum of immune cell phenotypes in fish since we found only a small number of cells with clear pro- or anti-inflammatory signatures.

## 1. Introduction

The innate immune system (IIS) serves as the first line of defense in multicellular animals [1]. IIS cells, such as macrophages and neutrophils, not only protect organisms from pathogens but also regulate development, homeostasis, and tissue repair [2]. Among multicellular animals, *Danio rerio* (zebrafish) continue to garner interest as a promising model organism for studying the properties of the immune system [3]. *Teleostei*, including zebrafish (at both adult and larval stages), have the potential to become key organisms for IIS-related studies, particularly with the growing interest in their single-cell transcriptomics [4]. Due to their similarity to mammals and ease of handling, zebrafish are widely used as models for human diseases but remain understudied as fish themselves [5]. The overall antiquity and evolutionary conservatism of the IIS make zebrafish a potentially powerful model for studying animal pathogenesis, injuries, and wound healing; macrophages and neutrophils are known to play a crucial role in these processes [6]. In addition, there is strong evidence highlighting the crucial role of zebrafish inflammatory pathways in developmental and aging processes [7].

The fish IIS comprises various cell types such as macrophages, neutrophils, mast cells, thrombocytes, dendritic cells, B cells, and rodlet cells. These diverse immune cells are synthesized in the fish kidney, which is considered a functional ortholog of mammalian bone marrow and serves as the main hematopoietic organ, producing almost all known immune cell types and performing both innate and trained immune responses [8,9]. A persistent challenge in immune system biology is the difficulty in defining and distinguishing different blood cell populations. There are numerous immune cell subtypes that flow and morph into one another, making them inseparable in vivo and irreproducible in vitro [10,11,12]. Therefore, single-cell transcriptomics data can serve as a powerful foundation for developing novel research strategies in fish immunology. In recent years, researchers have increasingly collected diverse transcriptomic single-cell data for fish, the combined analysis of which could help to identify key markers of different immune cells that are not dependent on the specifics of individual experiments but rather are inherent in the general population of these cell types [4]. A number of single-cell transcriptomic studies have identified key marker genes in various immune cells in zebrafish, emphasizing the important roles of macrophages and neutrophils in fish immunity [9,13,14,15]. The detailed molecular profiles of macrophages, neutrophils, and B cells in the zebrafish kidney were recently investigated [8]. Additionally, cross-organ scRNAseq experiments revealed highly conserved expression profiles of immune cells [16].

Summarizing the facts above, we already possess considerable detailed knowledge about the organization of the IIS in fish thanks to single-cell transcriptomics, although the depth of study varies. However, we are still far from having a comprehensive, systematic view of fish immunogenetics compared to the mammalian one. There are several dozen experiments available for adult and immature developmental stages of zebrafish and a few experiments for other fish species [4]. Individual scRNAseq datasets undoubtedly provide valuable insights into fish immunogenetics. However, simple summarizing of marker gene lists of various immune cells from different experiments is not an optimal strategy for analyzing such large and heterogeneous data. The proper meta-analysis of scRNAseq data could provide much better results through the integration and generalization of heterogeneous cellular samples obtained from fish with different genotypes, ages, and experimental characteristics. This approach could identify crucial genetic systems expressed in various immune cell subpopulations. There are many examples of productive applications of meta-analysis in the context of single-cell transcriptomics, including the study of leukocyte diversity in atherosclerotic mouse aortas [17], the identification of SARS-CoV-2 entry genes across tissues and demographics [18], and the study of responses of tumor-reactive CXCL13+ T cells to immune-checkpoint blockade [19]. However, for zebrafish, there is only one recent scRNAseq meta-analysis dedicated to the identification of genes involved in the heart regeneration process [20].

Therefore, conducting a meta-analysis of IIS-related scRNAseq datasets for zebrafish is an urgent task. In addition to the informative and predictive power of such meta-analysis, the identified genetic systems can be compared with those of mammals to investigate the degree of evolutionary conservatism of the IIS in *Vertebrata*. In such complex cases, modern approaches to the integration of transcriptome and interactome data can help in generating new knowledge and hypotheses about living systems [21]. Our work aims to bridge the current gap in understanding the core molecular genetic systems of macrophages and neutrophils in zebrafish compared to mammals. We identified novel marker genes for macrophages, neutrophils, and other cells of the IIS in zebrafish and transcription factors involved in myelopoiesis in fish. The discovered genes and pathways of fish neutrophils and macrophages require further study in the field of immunogenetics and transgenesis to evaluate their functions in pathogenesis, regeneration, and aging. Thus, our research could contribute to the development of novel applications of zebrafish as a model of human immunity. In addition, our work methodologically complements existing strategies for the meta-analysis of scRNAseq data. Our analysis strategy, which consisted of initial soft filtering of the data followed by more stringent filtering, proved to be flexible and effective, providing a diversity of cell types and high read coverage in the final dataset. We think that in future meta-analyses, researchers should select the threshold for filtering cells by the number of reads based on the quality of the original data rather than blindly using certain thresholds in data analysis pipelines. Therefore, the general steps of our meta-analysis can be used in future studies devoted to the joined analysis of scRNAseq data.

## 2. Materials and Methods

The overall pipeline of carried meta-analysis and post-analysis is presented in Figure 1. The main steps are described below, and in separate Section 2.1, Section 2.2, Section 2.3, Section 2.4 and Section 2.5, we explain them in detail.

1.For our meta-analysis, we searched and downloaded the suitable scRNAseq datasets described in our previous study [4] and experiments that were added to GEO NCBI during the last year;2.The downloaded datasets were converted to zebrafish gene identifiers (GRCz11). For each experimental dataset, we made a separate dataframe used to create the corresponding Seurat object. Seurat objects were annotated according to the general information for the experiments (GEO identifiers, fish age, and genotype);3.Individual Seurat objects went through a soft initial filtering procedure and were merged into a single Seurat. Afterward, to correct batch effects, we used normalization and harmonization procedures. In the resulting space, the main types of immune and non-immune cells of the zebrafish kidney were identified;4.Further filtering involved the removal of non-immune cells, immune cells with insufficient coverage, and the removal of genes not specific to immune cells. One experiment was also excluded from the final dataset because of the strong individual batch effect, which cannot be completely eliminated by Harmony and may lead to incorrect data integration;5.The final dataset was normalized and harmonized. In the resulting clusters, we identified immune cell types based on literature data, their marker genes, and TFs;6.The identified marker genes of different cell types were ranked and classified into one of four groups based on their expression level and the strength of upregulation in the target cell type;7.Next, for the most highly specific and highly expressed genes of macrophages and neutrophils, co-expression (Step 7 in Figure 1) and protein–protein interactions (Step 8 in Figure 1) were extracted, which allowed us to reconstruct gene networks relevant to the immune cells of these two types (Step 9 in Figure 1).

### 2.1. Data Collection

Following the search methodology from our earlier study [4], we selected relevant datasets that contained at least 5000 cells from zebrafish kidneys to ensure comparability in the meta-analysis. Thirteen experiments from the GEO NCBI database [28] met the inclusion criteria and were analyzed further: GSE100910 [29], GSE112438 [30], GSE130487 [31], GSE150373 and GSE176036 [32], GSE151231 [33], GSE166646 [34], GSE179401 [35], GSE190794 [36], GSE191029 [37], GSE242133 [8], GSE246039 [38], and GSE252788 [9]. Detailed information about these experiments is given in Table 1 and Appendix A.

For the further meta-analysis, we downloaded and used the count matrices or Seurat objects provided by the authors in the Appendix A in the corresponding GEO NCBI records (available by their GEO NCBI identifiers).

### 2.2. Data Meta-Analysis

#### 2.2.1. Software and Packages Used

All data analysis steps were performed using the R programming language v. 4.4.0 and the RStudio desktop environment v. 2023.12.1. The following packages were used: Seurat v. 5 [23], Harmony [24], Genekitr [22], hdWGCNA [26], tidyverse [39], dplyr, readr, ggplot2, cowplot, and patchwork. Bar plots and heatmaps were visualized using Python v. 3.11 and the Seaborn (function seaborn.heatmap) and Matplotlib (function matplotlib.pyplot.bar) packages. A Sankey diagram was created using the SankeyMATIC web service.

#### 2.2.2. Dataset Integration and Cell Type Identification

The initial step of our analysis involved merging individual sample count matrices from the selected experiments. Due to variations in gene identifiers across the analyzed experiments, the TransId function from the Genekitr package was used to convert all identifiers to an Ensembl format. Subsequently, individual count matrices were used to create Seurat objects via the CreateSeuratObject function from the Seurat package, applying soft filtering parameters: min.cells = 10, min.features = 200, and mitochondrial gene expression less than 10%. Multiple Seurat objects were then combined into a unified Seurat object using the merge function. Normalization and variance stabilization were performed using SCTransform v. 2. To initially estimate the number of clusters prior to harmonization, principal components were calculated (RunPCA function, npcs = 30, verbose = F), followed by the calculation of UMAP projections of cells (RunUMAP function, reduction = “pca”, dims = 1:30, verbose = FALSE). Preliminary clusters (prior to harmonization) were identified using the FindNeighbors (reduction = “pca”, dims = 1:30, verbose = F) and FindClusters (resolution = 1.0, verbose = FALSE) functions.

Subsequently, we used the Harmony algorithm to eliminate batch effects from individual datasets. The RunHarmony function was applied to the SCT assay of the unified Seurat object using the following parameters:group.by.vars = "orig.ident", project.dim = TRUE, plot_convergence = TRUE, verbose = TRUE, assay.use = "SCT".

After harmonization, Harmony components were utilized for uniform manifold approximation and projection (UMAP) calculation and cluster identification: RunUMAP, reduction = “harmony”, dims = 1:23. Clusters were then identified using the FindNeighbors (reduction = “harmony”, dims = 1:23) and FindClusters (resolution = 1, verbose = TRUE) functions. The cell types of these clusters were annotated based on marker genes from [8].

#### 2.2.3. Analysis of the Immune Cell Subset and Identification of Marker Genes

The subsequent analysis aimed to identify immune cell-specific genes that are highly expressed within immune cell types. To calculate immune-related marker genes, we used the FindMarkers function with the following parameters: only.pos = TRUE (upregulated), min.pct = 0.02 (expressed at least in 2 percent cells of the immune population), and logfc.threshold = 0.5 (upregulated by log2FC ≥ 0.5 in the immune population). Afterward, cells from non-immune clusters were excluded from the Seurat object. Further filtration was used to remove immune cells with low coverage. In particular, we selected immune cells with a high number of counts (3000–30,000 per cell), which expressed 1000–4000 genes at a non-zero level using the subset function. Further calculations and analyses were conducted on this most informative and immune cell-specific subspace (matrix consisting of selected immune cells × immune-related marker genes). Subsequently, we repeated the harmonization process on the immune subset using the RunHarmony function. The first 26 harmony dimensions were used for UMAP projection (RunUMAP) and cluster identification (FindNeighbors; FindClusters). Marker genes of individual clusters were calculated using the functions PrepSCTFindMarkers and FindAllMarkers: only.pos = TRUE (upregulated), min.pct = 0.25 (expressed at least in 25 percent cells of target clusters), and logfc.threshold = 0.5 (upregulated by log2FC ≥ 0.5 in target clusters compared to all other clusters).

To precisely investigate the cell types of the revealed immune cell subset, we compared expression within all Seurat clusters to assess their similarity. In particular, the function AverageExpression was used on an immune cell subset to calculate the expression matrix of all Seurat clusters in average counts-per-cell (ACPC) values (matrix of Seurat clusters × ACPC values of immune-related marker genes). This matrix was used to calculate the correlation matrix using the pandas.DataFrame.corr function. Based on the correlation matrix, we calculated the distance matrix (1 − abs(correlation)). For hierarchical clustering, the square form of the distance matrix was given to the function scipy.cluster.hierarchy.linkage with the following parameters: method = ‘ward’, metric = ‘Euclidean’, and optimal ordering = ‘True’. The majority of clusters were classified based on the expression of their marker genes [8] into neutrophils, macrophages, B cells, and hematopoietic stem cells. The minor cell types that we also detected were macrophage-like cells, neutrophil-like cells, immature neutrophils, monocytes, and T cells. After initialization of all cell types, cell-type-specific markers were calculated using the following functions FindMarkers: ident.1 = c(target cluster numbers), only.pos = TRUE (upregulated), min.pct = 0.25 (expressed at least in 25 percent cells of the target cell type), and logfc.threshold = 0.5 (upregulated by log2FC ≥ 0.5 in target cell type); ident.1 were different vectors corresponding to identified cell types that were compared with the rest of the immune clusters. Once we had identified markers in the cell type cluster groups, we identified transcription factors (TFs) among them using a list of transcription factors known for zebrafish in the database AnimalTFDB4 [25].

#### 2.2.4. Classification of Cell-Type-Specific Markers

All cell-type cluster markers were classified into four groups. The first group consisted of cell-type-specific genes with log2FC ≥ 2 and average counts-per-cell (ACPC) ≥ 2. This group included the most specific markers whose expression could be reliably detected in small datasets of immune cells. The second group included genes that were highly expressed in a given cell type with ACPC ≥ 2 but were less specific, with 0.5 ≤ log2FC ≤ 2. The third group consisted of cell-type-specific genes with log2FC ≥ 2 but low expression levels with ACPC ≤ 2. These cannot be fully considered marker genes, but their expression may be inducible under specific conditions and could play a role in the immune status of the cell. The fourth group comprised the remaining differential genes with low basal expression, where ACPC was ≤2 and with 0.5 ≤ log2FC ≤ 2.

### 2.3. Identification of Human Orthologs of Fish Immune Genes

All gene identifiers of detected marker genes from Groups 1–4 were used to search for corresponding orthological genes in the *H. sapiens* genome. The Ensembl Biomart data mining tool [40] was used, and lists of zebrafish genes were provided as input sources. The following search options were used: Database: Ensembl Genes 112; Dataset: Zebrafish Genes (GRCz11); Filters: Gene: Input external references ID list: Gene stable ID(s); Attributes: Homologues: Orthologues (F–J): Human Orthologues: Human gene name, human orthology confidence.

### 2.4. Reconstruction of Highly Expressed Macrophage and Neutrophil Gene Networks

The identified genes from Group 1 and Group 2 for macrophages and neutrophils were used to reconstruct cell-type-specific gene networks. The STRING-db database v. 12.0. was used as a source of interactions of studied protein-encoding genes [41]. Combined lists of macrophage-specific and neutrophil-specific genes were submitted to the web server version of the database using the option Search Multiple Proteins by Names/Identifiers. For the reconstruction of networks, we used the following parameters: minimum required interaction score: 0.4 (medium); sources of active interactions: experiments and databases; network display options: show your query protein names. The resulting protein–protein interaction networks were saved as a short tabular text output.

To search for additional associations between immune genes in zebrafish, we performed co-expression analysis on the final dataset of immune cells using the hdWGCNA [26] package. The analysis was performed according to a basic tutorial vignette, and its main steps consisted of setting up a Seurat object for hdWGCNA, the construction of metacells, and the construction of a co-expression network. After reconstruction of the co-expression matrix, we saved the topological overlap matrix and used the exportNetworkToCytoscape option of this matrix to export the resulting list of all edges in a Cytoscape-usable format. From the full co-expression edge list, co-expression interactions for macrophage-specific and neutrophil-specific genes with a co-expression threshold of at least 0.1 were extracted for further analysis.

Then, once we had data on protein–protein and co-expression interactions of neutrophil and macrophage genes, these edgelists were combined and used as input into the Cytoscape environment v. 3.10.2 [27] using option File–Import–Network from File. When importing edgelists, additional attributes were specified for interactions: co-expression or protein–protein interaction. Additional meta-information, such as gene expression levels, their annotations, and belonging to a certain group, was imported into the networks using the File–Import–Table from the File option. The layout of the networks was carried out according to the following principles: transcription factors were placed at the top of the network, known and candidate marker genes and their interactors were placed below TFs, and the remaining genes were placed below marker genes. Node degrees were estimated using the built-in NetworkAnalyzer tool. The final network layouts were saved and exported from the Cytoscape environment to Figures.

### 2.5. Functional Annotation

For gene-set enrichment analysis of genes of interest, the ShinyGO v. 0.8.0 tool was used. The following options were used: Species: Zebrafish genes GRCz11; FDR cut-off = 0.05; pathway database: KEGG; minimal pathway size = 2; remove redundancy: yes; abbreviate pathways: yes.

## 3. Results and Discussion

### 3.1. Zebrafish Kidney Marrow scRNA-Seq Dataset Integration and IIS Cell Classification

Based on the search methodology described in our prior study [4], we selected 13 relevant datasets consisting of at least 5000 cells from zebrafish kidneys [8,9,29,30,31,32,33,34,35,36,37,38]. The main steps of our initial analysis, filtration procedures, and cellular compositions of the final subset are presented in Figure 2. After applying soft filtration to 13 datasets, we obtained a total of 382,295 cells, which were subsequently normalized, harmonized, and clustered into 65 distinct cell groups (UMAP is shown in Figure 2B). Based on known gene markers for non-immune and immune cell types in the zebrafish kidney described in [8], we determined that 38 clusters correspond to immune cells, including 254,256 cells and representing five main cell types: hematopoietic stem cells (HSCs), B cells, macrophages, T cells, and neutrophils (showed as non-black-colored clusters in Figure 2B). The remaining 27 clusters correspond to non-immune cells, including 128,039 cells, and represent six main cell types: non-immune HSCs, erythrocyte progenitors, kidney mucin cells, distal tubule cells, proximal tubule cells, and multiciliated cells (showed as black-colored clusters in Figure 2B). The expression plots of the key marker genes for both immune and non-immune cells are provided in Appendix A.

To ensure a more precise and accurate final classification of immune cells, we filtered the final dataset in four steps. The first step involved filtering out cells from non-immune clusters. Afterward, we left in the resulting Seurat object only 3737 genes upregulated in the immune cells compared to non-immune ones (log2FC ≥0.5; see Appendix A). Then, due to the sufficient number of immune cells (254,256 cells in total), more stringent filtering was performed prior to the final data analysis. Specifically, cells with the best available read coverage, ranging from 3000 to 30,000 read counts per cell and expressing between 400 and 4000 unique genes per cell, were selected. The filtered dataset was composed of 96,509 immune cells. The different colors in Figure 2B) highlighted the presence of immune cells from initial clusters in this subset. During the subsequent step of harmonization and clustering of this subset, it was noted that the selected immune cells from the GSE112438 experiment revealed significant differences in embeddings compared to the cells from other experiments. This led to the exclusion of this experiment from the final analysis (see Appendix A). Consequently, an additional 2111 cells were excluded, resulting in a final dataset comprising 94,398 cells from 12 experiments and 3737 relevant immune genes (the expression matrix of the final dataset is provided in Appendix A). The cellular composition of the final dataset, resulting from the further integration of individual scRNA-seq experiments, is detailed in Figure 2C.

To sum up, our filtering approach allowed us to effectively select the highest quality cells available, which subsequently facilitated the accurate classification of the specific molecular genetic systems of individual cell types. This process reduced the representation of several cell clusters in the final dataset, marked in gray in Figure 2B, which appear to be in a transitional state among stem and mature immune cells, most likely neutrophils and T cells. Nevertheless, our final immune dataset consists of all terminal types characterized by pronounced genetic signatures in satisfactory quantities, marked in pink, or large quantities, marked in red or dark red, relative to the original dataset. Consequently, we maintained the balance between the qualitative composition and read coverage of individual cells. This filtering procedure suggests that researchers need to reasonably select thresholds when filtering scRNA-seq data, as these decisions can greatly affect the quality and results of the final analysis. Using the known markers [8], nine immune cell types were identified: hematopoietic stem cells, macrophages, neutrophils, B cells, T cells, monocytes, immature neutrophils, neutrophil-like cells, and macrophage-like cells (Figure 2C). The selected experiments are diverse in their design, including different mutant genotypes, various cell isolation and sorting protocols, and fish of different ages, as detailed in Appendix A. Therefore, it is not surprising that these experiments were quite heterogeneous in their cellular composition. Due to the inclusion of a sufficient number of experiments in our analysis, it was possible to compile a representative dataset balanced in the composition of individual cell types.

The overall information about the properties of the final dataset is presented in Figure 3. In particular, we used best-quality immune cells marked in red in Figure 3A. Classification of these immune cells revealed 44 clusters shown in Figure 3B, and corresponding immune cell types were revealed based on marker genes provided in [8]; the expression of certain important markers is shown in Figure 3D. The expression levels of the top 10 marker genes for each cell type are provided in Appendix A. To achieve a more precise definition of the cell types, hierarchical clustering was performed; the results are depicted in Figure 3C.

Thus, we revealed immune markers of the main IIS cell types and compared them with the study [8]. Five primary cell populations were clearly identified in our dataset: neutrophils, macrophages, hematopoietic stem cells, B cells, and immature neutrophils. Additionally, based on specific known markers, UMAP clustering, and hierarchical clustering, monocytes, immature T cells, macrophage-like, and neutrophil-like cells were identified. These cell types, their corresponding clusters, the number of cells in individual clusters, and the total number of cells by cell type are presented in Table 2. Detailed information about cell type markers and their classification is provided in Appendix A. In the sections below, we provide comprehensive information on the most relevant immune cell markers identified in our analysis.

#### 3.1.1. Hematopoietic Stem Cells

Hematopoietic stem cells are located in the top right area of the UMAP projection (Figure 3B, Table 2). HSCs are characterized by the increased expression of numerous ribosomal genes, including 17 *rpl* and 10 *rps* genes. Additionally, well-known marker genes such as *foxp4*, *rsl1d1*, *csf1rb*, *gtpbp4*, *rps7*, and *rpl7* exhibit strong upregulation in the cells of HSC clusters (Appendix A).

#### 3.1.2. Macrophages and Macrophage-like Cells

Macrophages are located in the middle right area of the UMAP projection (Figure 3B, Table 2). The expression of known macrophage markers, including *c1qc*, *c1qa*, *marco*, *c1qb*, *lgals3bp.1*, *hmox1a*, *fgl2a*, *grna*, *grna.1*, *ctss2.2*, *fthl27*, *ccl34a.4*, *s100a10b*, and *ctsd*, was confirmed. Macrophage-specific genes were strongly upregulated in clusters 2 and 4. Additionally, clusters 10, 32, and 40 showed significant similarity to the main clusters and were also identified as macrophages (Figure 3C).

Macrophage-like cells were identified as a highly heterogeneous group of small clusters occupying the bottom right area of the UMAP projection (Figure 3B). These cells exhibit high expression of several known macrophage markers, including *grna.1*, *grna.2*, *ccl34a.4*, *lgmn*, *s100a10b*, *ctss2.2*, and *ctsd*, but not classical markers of mature macrophages such as *marco*, *c1qa*, *c1qc*, and *c1qb*.

#### 3.1.3. Mature Neutrophils, Immature Neutrophils, and Neutrophil-like Cells

Neutrophils form a large group of cells located in the middle left area of the UMAP projection (Figure 3B). These cells are characterized by increased expression of markers such as *lyz*, *cpa5*, *sms*, *lect2.1*, *npsn*, *mpx*, *nccrp1*, *ncf1*, *il1b*, and *scpp8*. This group of cells is distinctly different from stem cells and other immune subtypes. Based on clustering (Figure 3C) and the expression of specific markers, such as *abcc13*, *hsd3b7*, and *si:ch1073-429i10.1* (Appendix A), ten clusters were classified as mature neutrophils (see Table 2). Additionally, two clusters, which are positioned closer to T cells and stem cells on the UMAP projection (Figure 3B), were identified as immature neutrophils. Among the highly expressed markers of immature neutrophils are well-known genes such as *ncf1*, *il1b*, *mmp9*, *scpp8*, *lect2.1*, *npsn*, *mpx*, and *nccrp1*. Compared to mature neutrophils, these cells additionally express genes such as *rgs2* and *gpr84* but do not express key markers of mature neutrophils, including *lyz*, *cpa5*, and *sms*. We also identified neutrophil-like cells in the bottom right of the UMAP projection that express certain neutrophil-specific genes, including *lect2.1*, *npsn*, *cpa5*, *mpx*, and *sms*, but at lower levels than classical neutrophils (Appendix A).

#### 3.1.4. B Cells, T Cells, and Monocytes

B cells occupy a compact area in the upper left of the UMAP projection (Figure 3B). They express many known markers, such as *igic1s1*, *plaat1l*, *cxcr4a*, *bhlhe40*, *bcl2l10*, *irf8*, *ccr9a*, and *cd74a*. Interestingly, these cells also exhibit high levels of several *rpl* and *rps* genes, similar to stem cells, indicating the occurrence of translation processes within this cell population.

T cells were identified by the expression of the *runx3* gene and the lymphocyte-specific gene *ccr9a*. These cells are most likely immature T cells, as they represent a minor component of kidney immune cells and, like B cells, express numerous genes related to translation processes.

Monocytes were identified based on the primary cell marker *cd34*, which corresponds to the monocytic lineage of progenitors [42]. This extensive group of cells is located between stem cells and macrophages, retaining some progenitor properties and characterized by the elevated expression of six *rpl* genes and five *rps* genes. The corresponding plots are provided in Appendix A.

### 3.2. Detailed Analysis of the Molecular Genetic Systems of Immune Cells

During our analysis, we revealed that macrophages and neutrophils are the two most terminal and distinctive cell types presented in our datasets, and they are enriched in the expression of many specific immune genes. We also revealed minor subpopulations of B cells and immature T cells, confirming the hypothesis that the zebrafish kidney could serve as a potential source of lymphocytes [8]. The other cell types either show similarities to macrophages and neutrophils or exhibit significant stemness and do not express specific immune genes as strongly. Hence, in the following sections, we primarily focused on the in-depth analysis of the molecular–genetic systems of macrophages and neutrophils.

For further analysis, we classified immune markers into four groups by their average expression in identified cell types and by their upregulation folds. This classification is described in Materials and Methods, Section 2.2.4. In short, genes in Group 1 (cell-type-specific highly expressed genes: log2FC ≥ 2; ACPC ≥ 2) and Group 2 (highly expressed genes: ACPC ≥ 2; 0.5 ≤ log2FC ≤ 2) should be considered the highest priority for further experimental and theoretical research. The roles of the markers from Group 3 (cell-type-specific but weakly expressed) and Group 4 (less specific and weakly expressed) require additional meta-analyses or experiments to verify their involvement in myelopoiesis processes in zebrafish. The expression heatmap of identified cell-type-specific highly expressed genes (Group 1) in all Seurat clusters is depicted in Figure 4.

We identified 62 highly specific marker genes for macrophages and macrophage-like cells (see Figure 4A), thirteen of which were previously highlighted in the list of the 20 most specific macrophage marker genes [8]. For mature neutrophils and immature neutrophils, 58 specific genes were revealed, depicted in Figure 4B; among them were thirteen of the most relevant neutrophil genes mentioned previously [8]. B-cells and HSCs can be identified by their specific twelve and five markers, respectively, as shown in Figure 4C. Seven B-cell markers were previously reported by Hu et al., as well as two markers of HSCs, which were validated in our analysis [8]. Therefore, our empirical classification is reliable since we used rather stringent thresholds for both fold-changes and basal expression levels for selecting these genes. Furthermore, our classification and selection of these genes is strongly supported by previous data [8] and expands key marker genes for neutrophils and macrophages in the zebrafish kidney.

Thus, the number of newly discovered key markers of immune cells in the zebrafish kidney does not exceed one hundred genes, most of which are expressed in macrophages and neutrophils; all marker genes and their classification are given in Appendix A. Therefore, in Section 3.2.1, Section 3.2.2 and Section 3.2.3, we discuss in detail the potential roles of our discovered genes in fish immune cells based on the available literature on different fish species, mice, and humans. In Section 3.2.4, we review and discuss, in the context of recent discoveries, our newly discovered transcription factors involved in myelopoiesis in zebrafish. Throughout Section 3.2.1, Section 3.2.2, Section 3.2.3 and Section 3.2.4, we used the generally accepted gene nomenclature spelling: human gene names are marked by upper case (*NAME*), mouse gene names are marked by title case (*Name*), and fish gene names are marked by lower case (*name*).

#### 3.2.1. The Novel Marker Genes of Zebrafish Macrophages

For fish macrophages (Figure 4A), our meta-analysis verified the expression of 13 out of the 20 primary kidney marrow macrophage marker genes from the study [8]. Additionally, the expression of 41 newly identified marker genes was confirmed, including six cathepsin genes (*ctsa*, *ctsba*, *ctsc*, *ctsh*, *ctsk*, and *ctsz*), three annexin genes (*anxa1a*, *anxa3b*, and *anxa5b*), two cd74 genes (*cd74a* and *cd74b*), two keratin genes (*krt4* and *krt18b*), two solute carrier family genes (*slc2a6* and *slc22a21*), and two major histocompatibility complex class II genes (*mhc2a* and *mhc2dab*). Cathepsins are essential as a defense against intracellular bacteria. The upregulation of cathepsins *CTSB*, *CTSC*, *CTSH*, and *CTSZ* is a marker of the M1 phenotype of human macrophages, while the upregulation of *CTSK* is more pronounced in M2 macrophages [43]. In mice, annexin *Anxa1* has been shown to induce macrophage migration to participate in muscle regeneration through the AMPK cascade [44], annexin *Anxa3* upregulates the infiltrated neutrophil-lymphocyte ratio to remodel the immune microenvironment in hepatocellular carcinoma [45], and annexin *Anxa5* regulates the polarization of liver macrophages by acting on its primary target, pyruvate kinase M2 [46]. Increased *CD74* gene expression in atherosclerosis is an indicator of macrophage inflammatory responses [47]. A cellular model of head and neck squamous cell carcinoma revealed the high correlation between the expression levels of keratin genes (*KRT4*, *KRT24*, *KRT78*, and *KRT13*) and the infiltration levels of CD8+ T cells and macrophages [48]. One of the solute carrier family proteins in mice, *Slc37a2*, has been shown to suppress macrophage inflammation through the regulation of glycolysis [49]. Among the newly identified macrophage marker genes, four genes encoding receptors were found: *havcr1* (hepatitis A virus cellular receptor 1), *mrc1b* (mannose receptor C type 1b), which is upregulated in M2 macrophages in fish [50], prosaposin *psap*, and *pglyrp5* (peptidoglycan recognition protein 5), which has been shown to play a role in antimicrobial activity and enhancing macrophage phagocytosis in *Larimichthys crocea* [51]. Additionally, four transcription factors were identified: *junba*, which is upregulated during zebrafish heart regeneration [52], *mafba*, known for its role in regulating microglia colonization in zebrafish [53], *tcima*, recently identified as a macrophage-enriched gene in zebrafish [54], and *id1*, previously identified as relevant to NK cells in zebrafish [15]. Among other relevant genes, the *AHNAK* nucleoprotein has been observed to suppress tumor proliferation [55], while *Cfp* (complement factor properdin) knockout mice exhibited a shift towards M2 macrophage polarization [56] and galectin *lgals2a*, whose ortholog in an in vivo mouse study demonstrated an increase in macrophage numbers [57]. Additionally, we detected the strong expression of lysozyme g *lygl1*, whose expression in fish immune cells was confirmed in a study [58], indicating membrane-spanning 4-domains member *ms4a17a.10*, previously shown to be expressed in fish B cells [15], surfactant protein Bb *sftpbb*, which was confirmed to be upregulated in zebrafish macrophages [59], syndecan 4 (*sdc4*), whose ortholog knockdown in mice (*Sdc4*) leads to a pro-inflammatory phenotype in macrophages and the development of atherosclerosis [60], vacuole membrane protein 1 (*vmp1*), whose ortholog knockout (*VMP1*) in human cell lines triggered the release of pro-inflammatory cytokines [61], and gene *zgc:174904*, whose elevated expression was previously detected in fish B cells [62]. The roles of the genes discussed above are confirmed to varying degrees in macrophages and, in some cases, in other immune cells of zebrafish or only for mammals. However, for five of the newly identified genes, there is currently no information about their specific roles in macrophages: brain-subtype creatine kinase *ckbb*, chloride intracellular channel protein *clic2*, phorbol-12-myristate-13-acetate-induced protein 1 *pmaip1*, tetraspanin-36 (*tspan36*), and tubulin polymerization promoting protein (*LOC100333482*).

Macrophage-like cells (Figure 4A) express a set of genes typical for the classical macrophage population, including *lgals2a*, *lygl1*, *cd74a*, and *cd74b*, albeit at lower levels. In addition to the known markers *grna.1*, *grna.2*, *cxcl12a*, and *lgmn*, we identified higher expression of new potential markers in these cells, such as beta-microseminoprotein-like (*LOC110437731*), mtp family protein (*LOC100151049*), uncharacterized protein *LOC101883708*, and *si:ch211-194m7.4*. The beta-microseminoprotein-like protein has demonstrated antibacterial activity in chicken eggs [63], while the other genes remain largely uncharacterized.

#### 3.2.2. The Novel Marker Genes of Zebrafish Neutrophils and Immature Neutrophils

The expression of genes typical for mature neutrophils is illustrated in the top part of Figure 4B. Newly identified proteins include two ATP-binding cassette proteins (*abcb9* and *abcc13*), cathepsin S (*ctss1*), 15-hydroxyprostaglandin dehydrogenase (*hpgd*), 3′-phosphoadenosine 5′-phosphosulfate synthase 2b (*papss2b*), arachidonate 5-lipoxygenase-activating protein (*alox5ap*), cholesterol 25-hydroxylase-like protein 2 (*ch25hl2*), hydroxy-delta-5-steroid dehydrogenase (*hsd3b7*), phospholipid phosphatase 1 (*plpp1a*), plac8 onzin-related protein 6 (*ponzr6*), protein kinase C delta type (*prkcda*), S-adenosylmethionine decarboxylase proenzyme (*amd1*), serglycin (*srgn*), sulfotransferase (*sult2st1*), thy-1 cell surface antigen (*thy1*), very-long-chain (3R)-3-hydroxyacyl-CoA dehydratase (*hacd4*), vesicle-associated membrane protein 8 (*vamp8*), and a gene with predicted oxidoreductase activity *si:ch1073-429i10.1*. The expression of *abcb9* has been validated in zebrafish neutrophils [64]. Studies using mouse models have shown that knockout of cathepsin *Ctss* reduces neutrophil recruitment [65]. Inhibition of the *Hpgd* gene has been found to enhance neutrophil regeneration in mice [66], indicating its potential as a marker for neutrophil maturation. The *ALOX5AP* gene plays a role in neutrophil ferroptosis and compromises cell wall integrity during intracranial aneurysms [67]. In humans, *HSD3B7* gene expression is positively correlated with the infiltration of immune cells, including neutrophils [68]. The *ponzr6* gene has been previously confirmed in zebrafish neutrophils [69], while the study [70] demonstrated the high specificity of *prkcda* expression in fish neutrophils. The role of *amd1* in neutrophil maturation is noted in the research [9]. In tumor tissues, the suppression of *SRGN* gene expression correlated with lower immune infiltration of neutrophils and other immune cells [71]. In mice, cholesterol sulfate produced by the sulfotransferase *Sult2b1* acts on inflammatory neutrophils and prevents excessive intestinal inflammation [72]. *Thy-1* (*CD90*) is involved in the skin wound healing mechanism by activating various immune cell types in mammals [73]. The role of the *Hacd4* gene in inflammatory processes has been confirmed in mouse models [74]. Different Vamp genes in mice are engaged in neutrophil exocytosis, suggesting a similar function for the *vamp8* gene in fish [75]. The functions of the identified genes *abcc13*, *ch25hl2*, and *plpp1a* in zebrafish neutrophils need to be investigated in future research.

Similar to mature neutrophils, immature neutrophils (Figure 4B, middle section) highly express several established markers, such as *lyz*, *cpa5*, *lect2.1*, *npsn*, *mpx*, and *srgn*. We identified a distinct set of genes specific to immature neutrophils: disintegrin and metalloproteinase domain 8a (*adam8a*), activating transcription factor 3 (*atf3*), arginase (*arg2*), immune-related lectin-like receptor 4 (*illr4*), interleukin 6 receptor (*il6r*), MCL1 apoptosis regulator (*mcl1b*), ornithine decarboxylase 1 (*odc1*), purine nucleoside phosphorylase (*pnp5a*), a member of the RAS oncogene family rab44, scinderin-like b (*scinlb*), serine/threonine kinase 17B (*stk17b*), solute carrier family 7 member 3a (*slc7a3a*), sphingosine-1-phosphate receptor 4 (*s1pr4*), suppressor of cytokine-signaling 3a (*socs3a*), TIMP metallopeptidase inhibitor 2b (*timp2b*), transcobalamin beta b (*tcnbb*), tumor necrosis factor b (*tnfb*), v-set immunoregulatory receptor (*vsir*), *zgc:158343*, *si:ch211-264f5.2*, and BI1 family proteins *si:ch211-284o19.8*, *si:ch73-248e21.5*, *si:ch73-343l4.8*, *si:dkey-195m11.11*, *si:dkey-27h10.2*, and *zgc:158343*. The deletion or inhibition of the *Adam8* gene leads to impaired neutrophil transmigration in mice [76]. Atf3-deficient mice show weakened antibacterial immunity due to their involvement in the modulation of macrophage killing and migration functions [77]. The *ARG2* gene is a signature of N2 neutrophils in a rosacea-like model and is induced by LPS+ IFN-γ and IL-4 [78]. The *illr4* gene has been identified as a chemotaxis gene for neutrophils in fish [79]. It is known that human neutrophils produce interleukin 6 and its receptor *IL6R* during the inflammatory response [80]. While the anti-apoptotic gene *mcl1b* was previously recognized for its role in neutrophil maturation, our analysis indicates a more stable expression in immature neutrophil clusters [9]. Deletion of the *Odc1* gene in mice resulted in delayed neutrophil clearance and a reduction in the proresolving cytokine interleukin-10 [81]. The fibrinogen and endopeptidase activity of the *pnp5a* gene in fish during the inflammation and immune response has been confirmed [82]. *Rab44* gene expression has been shown to decrease during neutrophil maturation in the mouse model [83], a finding that our study supports for its ortholog in zebrafish. The proteomic analysis of zebrafish kidney neutrophils showed a significant increase in the expression of several proteins, including *scinlb*, in response to chemically-induced inflammation [84]. The *Stk17b* gene in mice and *STK17B* in humans have been identified as signatures of activated neutrophils in metastatic tumors [85]. The *slc7a3a* gene regulates lipid metabolism through the mediation of arginine-dependent nitric oxide synthesis in zebrafish cells [86], and nitric oxide subsequently modulates numerous physiological properties of neutrophils in response to various pathogenic stimuli, which may be a crucial feature of immature neutrophils, reflecting their functional adaptability [87]. The *S1PR4* gene in humans and the *S1pr4* gene in mice are known participants in immunity modulation, the regulation of homeostasis, and neutrophil migration [88]. In mice, the *Socs3* gene serves as a negative regulator of *Tbk1* and the interferon pathway and is expressed in neutrophils in response to the granulocyte–macrophage colony-stimulating factor combined with Prostaglandin E2, supporting the anti-inflammatory phenotype of neutrophils [89] and potentially maintaining their immature characteristics. The expression of *timp2b* could indicate its critical role in the development of local inflammation, resolution, and tissue repair in zebrafish, which is consistent with the revealed role of its ortholog *TIMP-2* in human leukocytes [90]. The *tcnbb* gene’s expression in neutrophils was confirmed in [8]. It is well-known that an injection of tnf-alpha precursor in zebrafish results in induced neutrophil recruitment [91], and *tnfb* may similarly stimulate immature neutrophils. Fish orthologs of the *vsir* gene have been confirmed in neutrophils and other immune cells in both humans and mice [92]. Huang’s dissertation [93] reports on the upregulation of the zgc:158343 gene during bacterial infection in zebrafish, and the upregulation of the *si:ch211-284o19.8* gene was revealed in the fish’s response to bacteria [94].

Neutrophil-like cells (Figure 4B, lower section) exhibit high expression levels of several neutrophil-related genes, including *lect2.1*, *npsn*, *mpx*, *srgn*, *ncf1*, *scinlb*, and *illr4*. The genes specific to these cells are *capzb*, *LOC103910140*, *si:ch211-9d9.1*, and *si:ch211-223l2.4*. The upregulation of *CAPZB* gene expression has been detected in human B cells in the context of lymphoma [95]. The expression of the glycoprotein-like gene *cd59* (*LOC103910140*) in fish is downregulated in response to alginate oligosaccharides [96], which similarly leads to reduced neutrophil infiltration and lower levels of inflammatory markers in mice [97].

#### 3.2.3. The Novel Markers of Other Cell Types

In B cells (Figure 4C, upper section), we detected the expression of *dusp2*, *ccl35.1*, *syk*, *grap2b*, and *kctd5a*. The *Dusp* genes, including *Dusp2*, function as regulators and specifiers of innate immune responses in mice, with their expression being controlled via the MAPK pathway following the binding of foreign agents to pattern recognition receptors [98]. High expression of *ccl35.1* has been identified in zebrafish dendritic cells [16]. The *Syk* gene has been shown to play an essential role in B cell activation in mice [99]. The downregulation of the *Grap2* gene was observed in the transcriptome of B cells from pregnant mice [100]. In the tumor microenvironment, *KCTD5* gene expression was positively correlated with macrophages and T cells while negatively correlated with plasma cells, mast cells, and memory B cells [101].

Increased expression of the *cd34* marker gene was observed in some monocytic clusters and HSCs (Figure 4C, middle section). Sialomucin (CD34) is known as a marker for the early maturity stages of human monocytic precursors, suggesting that the monocytic cells detected in our dataset may possess a high degree of immaturity and stemness [102].

We found that in HSCs (Figure 4C, lower section), in addition to the well-known *csf1rb* and *nanos1*, the genes *hist2h2l*, *myca*, and *si:ch211-161c3.6* are highly expressed. During *Aeromonas salmonicida* infection in *Cyprinus carpio*, the expression of *hist2h2l* significantly decreases, which may indicate the mobilization of stem cell differentiation in the immune system [103]. The role of the cell cycle regulator *myca* in the proliferation of stem cells in zebrafish was demonstrated in [104]. The transcription factor *si:ch211-161c3.6* was previously identified as a marker of bone marrow stem cells in fish [36].

#### 3.2.4. The Novel Transcription Factors Involved in Zebrafish Myelopoesis

Some of the specific genes identified as markers for immune cell types in our study, including several transcription factors (see Figure 5), are discussed in the context of myelopoiesis in fish across multiple studies. The review [105] covers various genes that regulate neutrophil homeostasis, including transcription factors *cebpa*, *cebpb*, *stat1a*, *myca*, and *spi1b*, which were confirmed in our analysis. In the study of the transcriptional network of the *zbtb14* factor, which regulates the development of monocytes and macrophages, factors *cebpa*, *myca*, *spi1b*, *irf8*, and *mafba* were found to be involved [106]. The genes *spi1a*, *mafba*, *mafbb*, *irf8*, and *hif1ab* identified in our analysis are potential immunotherapeutic targets for myeloid cells involved in neurovascular repair following ischemic stroke [107]. Transcription factors *klf6a*, *cebpb*, *cebpa*, *hif1ab*, and *spi1b* are highlighted as key players in granulopoiesis in fish [108]. These eleven factors are represented in Figure 5. Additionally, we identified 27 transcription factors to be involved in fish myelopoiesis and discussed them below.

According to our data, three of the factors broadly mentioned in the literature, *cebpa*, *cebpb*, and *spi1b*, are common to most types of immune cells. We identified that transcription factors *ybx1*, *zgc:162730*, *xbp1*, and *fosab* are also ubiquitously expressed in all immune cell types. The *YBX1* gene is known for its association with the immune system and its pro-oncogenic properties in human hepatocellular carcinoma [109]. Upregulation of *xbp1* expression has been observed in response to INH-lipopolysaccharide in zebrafish [110]. The *fosab* gene expression is upregulated under inflammatory conditions induced by Bisphenol AP in zebrafish [111].

A total of 17 transcription factors exhibited specific upregulation in the macrophage group (Figure 5, second section from the top), including five that are known to play key roles in monocyte maturation: *klf6a*, *spi1a*, *mafba*, *irf8*, and *mafbb*. The other 12 factors are *junba* (discussed above), *csde1*, *nr4a1*, *fosl2*, *atf4b*, *atf4a*, *nfe2l2a*, *id1*, *jdp2b*, *klf2a*, *zfp36l1b*, and *nr1h3*. The human *CSDE1* gene is implicated in the immune recognition of nascent tumorigenic cells [112]. In mice, *Nr4a1* deletion supports pro-angiogenic macrophage polarization and is essential for the maturation of inflammatory monocytes [113]. *FOSL2* contributes to M2 macrophage polarization and enhances the tumorigenic capacity of glioblastoma cells [114]. The transcription factor *ATF4* enhances M2 macrophage infiltration through the CCL2-dependent pathway [115]. *Nfe2l2* plays a role in preventing the upregulation of pro-inflammatory cytokine expression and suppressing the pro-inflammatory phenotype of macrophages in mice [116]. Macrophages expressing the *ID1* transcription factor have been shown to limit T cell infiltration in colorectal cancer [117]. Neomycin exposure has been shown to lead to the sustained activation of several genes in zebrafish macrophages, including *jdp2b*, which collectively mediate a switch to an anti-inflammatory state [118]. The expression of miR-2188-5p triggers the degradation of *klf2a*, leading to macrophage differentiation in *Ctenopharyngodon idellus* [119], suggesting that *klf2a* may support macrophage stemness. The anti-inflammatory properties of mouse macrophages are supported by the transcription factor *Zfp36l1*, whose expression significantly increases upon LPS treatment [120]. The *NR1H3* gene is associated with a pro-inflammatory phenotype in human macrophages [121].

The next group of TFs identified by our meta-analysis is predominantly expressed in neutrophils: *atf3*, *plek*, *id3*, *lrrfip1a*, and *raraa* (Figure 5, second section from the bottom). An increase in the expression of *PLEK* along with genes associated with neutrophil activation and degranulation has been observed in response to osteonecrosis of the femoral head [122], suggesting a potential role for this TF in neutrophil activation in fish as well. Although the role of *id3* in neutrophils remains unclear, it is known that exogenous supplementation with Id2 protein in mice can effectively protect them from colitis and inhibit NF-kB activation in neutrophils [123]. In human cells, *LRRFIP1* has been shown to inhibit the NLRP1 inflammasome and regulate hematopoiesis [124]. The elevated expression of *raraa* in zebrafish neutrophils has also been confirmed in [125], and the expression of the *Rarα* gene is associated with neutrophil differentiation in mice [126].

The lower section of Figure 5 highlights additional marker TFs expressed in various cell types, including *si:ch211-9d9.1*, which is expressed in neutrophil-like cells; *bhlhe40*, found in B cells; *foxp4* and *si:ch211-161c3.6*, both expressed in HSCs; and *mych*, which is present in both HSCs and B cells. It has been shown that *Bhlhe40* plays a key role in activated mouse B cells by restraining the germinal center reaction and preventing lymphomagenesis [127]. The gene *Foxp4* has been associated with regulating robust recall responses in mouse T lymphocytes [128]. The expression of *mych* was previously identified in zebrafish neutrophils, where it plays a role in inflammation and the formation of mycobacterial granulomas [129].

Thus, among identified genes and TFs, most have been functionally validated in various immune roles in mammals, which likely accounts for their detection as immune markers in zebrafish. Additionally, some genes already known to be specific to fish immune cells from the study [8] and TFs from four other studies [105,106,107,108] have been confirmed. However, these scattered data do not yet allow us to piece together a coherent picture of myelopoiesis in fish, indicating a need for further investigation. The newly identified candidate transcription factors for macrophages and neutrophils in fish suggest complex regulation of these cell types and their diverse functional roles. Therefore, in the subsequent section of the results, we focus on the reconstruction of gene networks consisting of highly expressed marker genes of neutrophils and macrophages, as well as transcription factors identified in our study. Using available data on protein–protein and co-expression interactions, we aim to visualize their molecular–genetic systems and characterize their functional roles.

### 3.3. Gene Networks and Functional Annotations of Key Macrophage and Neutrophil Marker Genes Identified through Integration of Zebrafish Kidney scRNA-Seq Datasets

During our meta-analysis, we encountered two main issues:1.Initial low coverage: The limited coverage of our data analysis was due to the inherent limitations of scRNA-seq technology (see Section 3.1). This limitation was partially addressed through a filtering procedure that allowed us to exclude cells with a minimal number of reads.2.Lack of precise information: There was insufficient detailed information on the immune properties and functions of individual genes in the identified immune cell type markers (see Section 3.2). This gap is often addressed by utilizing data from specific studies that examine the function of particular orthologous genes in humans or mouse models.

Given these challenges, we believe that the most effective strategy for bioinformatics analysis and interpretation of such data is to reconstruct gene networks of two major cell types identified during our meta-analysis (macrophages and neutrophils). This approach can provide a valuable blueprint for understanding the genetic systems involved in the peculiarities and regulation of macrophages and neutrophils. Combining data on protein–protein and co-expression interactions for macrophage-specific and neutrophil-specific genes (see Appendix A) from Group 1 (cell-type-specific highly expressed genes) and Group 2 (highly expressed genes), we reconstructed gene networks for highly expressed macrophage genes (Figure 6) and neutrophil genes (Figure 7). Genes without any protein–protein or co-expression associations were excluded from these networks.

#### 3.3.1. Macrophage-Specific Gene Network

The resulting gene network of highly expressed macrophage genes (Figure 6) comprises 113 protein-coding genes connected through 764 interactions, of which 95 are protein–protein interactions, and 669 are co-expression associations. The Cytoscape session file (.cys) with this network is provided in Appendix A. The network includes 13 transcription factors (highlighted at the top of the figure as hexagons). The transcription factors *spi1a*, *cebpb*, and *mafba* are already known for their roles in myelopoiesis [105,106,107,108]. We predicted the involvement of ten new potential regulators of macrophage gene expression: *nr4a1*, *atf4a*, *fosl2*, *junba*, *fosab*, *xbp1*, *nfe2l2a*, *ybx1*, *nme2b.1*, and *id1*. Collectively, the transcription factors are associated with 50 potential target genes through five protein–protein and 71 co-expression interactions.

It is worth noting that the newly predicted factors *fosab* and *fosl2* were found to have protein–protein interactions with the already known important regulator of fish myelopoiesis, *mafba*, which, in turn, interacts with the candidate factors *atf4a* and *junba*. Additionally, the transcription factor *cebpb*, noted to be involved in myleopoiesis, has a protein–protein interaction with *atf4a*. Taken together, such interactomic data may hint at the complex nature of the regulation of maturation and polarization of zebrafish macrophages, which are able to restructure their metabolic properties depending on their spatial context [16] and the presence of foreign agents, wounds, and other factors [15].

Among the genes presented in the macrophage-specific network, there are three small clusters highly enriched with internal protein–protein interactions:1.Twelve genes are related to immune functions. Six cathepsins: *ctsa*, *ctsba*, *ctsc*, *ctsh*, *ctsk*, and *ctsz*; four genes encoding major histocompatibility complex class II proteins (*si:busm1-266f07.2* (*mhc2a*), *mhc2dab*, *cd74a*, and *cd74b*); peptidoglycan recognition protein 5 (*pglyrp5*) and macrophage migration inhibitory factor (*mif*);2.Eight genes encode integral membrane proteins. Three genes of transporters of proteins into the endoplasmic reticulum (*sec61a1*, *sec61b*, and *sec61g*); three genes of signal sequence receptors (*ssr2*, *ssr3*, and *ssr4*); transmembrane protein 258 (*tmem258*); a subunit of the oligosaccharyl transferase complex *dad1*;3.Six genes of proton-transporting V-type ATPase complex: *atp6ap2*, *atp6v0e1*, *atp6v0ca*, *atp6v1g1*, *atp6v1e1b*, and *rnaseka*.

The network includes 15 out of the 20 key macrophage markers identified in the study [8], which are marked by ellipses outlined in black. Additionally, the network encompasses 34 potential macrophage markers, highlighted by ellipses outlined in blue. The most highly connected nodes in the network with more than five protein–protein interactions are *cd74a*, *ctsc*, *ctsh*, *ctsk*, *mhc2dab*, *sec61a1*, *sec61b*, *sec61g*, *ssr2*, and *ssr3*, while the most co-expressed genes with more than 35 associations include *atp6v0ca*, *c1qc*, *cfp*, *ctsa*, *ctsba*, *ctss2.2*, *fgl2a*, *grna*, *mafba*, *ms4a17a.10*, *psap*, and *si:dkey-5n18.1*. Genes with highest expression in network (log2(ACPC) >5) are *nme2b.1*, *s100a10b*, *grna.1*, *fthl27*, *cd74a*, and *hspa8*. Therefore, the significant enrichment of protein–protein interactions is primarily observed due to associations between protein-coding genes rather than transcription factor–gene interactions, highlighting the need for further research into the regulatory mechanisms of myelopoiesis and the specific roles of individual transcription factors within this network.

The identified co-expression patterns may provide insights into potential targets for specific transcription factors, which are detailed in Appendix A. Among the transcription factors, *mafba* exhibits the highest number of co-expression associations within the network, with 48 interactions, followed by *id1* with 17 connections. Additionally, the transcription factor *xbp1* engages in four protein–protein interactions with *cd74a*, *sec61a1*, *sec61b*, and *sec61g*.

#### 3.3.2. Neutrophil-Specific Gene Network

The reconstructed gene network of highly expressed neutrophil genes (Figure 7) encompasses 124 protein-coding genes with 2436 interactions, 67 of which are protein–protein interactions, and 2369 are co-expression interactions. The Cytoscape session file (.cys) with this network is provided in Appendix A. The network includes 13 transcription factors, five of which are known: *cebpa*, *cebpb*, *hif1ab*, *spi1b*, and *stat1a* [105,106,107,108], and nine newly predicted factors: *atf3*, *atf4b*, *id3*, *lrrfip1a*, *nme2b.1*, *plek*, *xbp1*, and *ybx1*. The factors *atf4b* and *cebpb* are linked by a protein–protein interaction, while *nme2b.1* and *ybx1* are co-expressed in neutrophils. The identified transcription factors are connected to 79 potential targets through four protein–protein and 201 co-expression interactions. The network includes 11 of the 20 key neutrophil markers from [8], highlighted by black-bordered ellipses. Additionally, 18 potential neutrophil markers, indicated by blue-bordered ellipses, are present in the network. The genes with more than five protein–protein interactions in this network are *actb1*, *actb2*, *actr2a*, *arpc1b*, *arpc3*, *arpc4l*, and *arpc5b*, while the genes with more than 85 co-expression associations are *scpp8*, *alox5ap*, *si:ch73-343l4.8*, *ppdpfa*, *si:ch211-284o19.8*, *illr4*, *adam8a*, *mapre1a*, *ctss1*, *vamp8*, *hacd4*, *lta4h*, and *mpx*. The most highly expressed genes in the network (log2(ACPC) >5) are *lyz*, *lect2.1*, *actb2*, *tmsb4x*, *pfn1*, *npsn*, *srgn*, *icn*, *actb1*, and *cpa5*.

Neutrophils demonstrate more pronounced overall co-expression compared to macrophages, which might indirectly suggest their more deterministic transcriptional regulation compared to macrophages. On the other hand, the presence of only one protein–protein interaction between their TFs (*cebpb*-*atf4b*) may speak in favor of their weaker co-regulation, which could mediate their specific pattern of target gene regulation.

In this neutrophil-specific network, there are 21 genes with a large number of internal protein–protein interactions:1.Two genes encoding actin (*actb1*, *actb2*) and five genes of actin-related protein 2/3 complex subunit: *actr2a*, *arpc1b*, *arpc3*, *arpc4l*, and *arpc5b*;2.Six genes related to actin cytoskeleton organization: plastin-2 *lcp1*, cofilin 1-like *cfl1l*, profilin *pfn1*, thymosin beta *tmsb4x*, myosin heavy chain 9a *myh9a*, and tropomyosin 1 alpha *tpm1*;3.Four genes involved in glycolytic process: enolase 1a *eno1a*, enolase 3 *eno3*, glyceraldehyde-3-phosphate dehydrogenase 2 *gapdhs*, and glucose-6-phosphate isomerase *gpia*;4.Glutathione reductase *gsr*, myosin light chain 12 *myl12.1*, transketolase b *tktb*, and *zgc:153867*.

The interactions of neutrophil transcription factors and their potential targets are detailed in Appendix A. The transcription factors *plek* (47 connections), *id3* (41 connections), *atf3* (32 connections), *lrrfip1a* (28 connections), *spi1b* (19 connections), and *xbp1* (16 connections) exhibit the highest number of co-expression associations with genes in the network.

#### 3.3.3. Functional Annotation of Highly Expressed Genes in Zebrafish Macrophages and Neutrophils

For macrophage and neutrophil marker genes from Group 1 and Group 2, we identified 15 and 25 enriched KEGG pathway terms, respectively. Seven of these pathways are shared in both cell types, though they often involve different genes. A comprehensive list of all enriched pathways and associated genes can be found in Appendix A, with key pathways highlighted in Figure 8 below.

Macrophage cells show significant enrichment in pathways such as “Protein processing in the endoplasmic reticulum”, “Protein export”, and “RNA degradation”, which are essential for their functional roles [130,131]. These cells also display specific enrichment in metabolic pathway components, indicating high metabolic activity and suggesting activation or polarization processes similar to those observed in humans. For instance, the metabolic pathways of glycine, serine, threonine, and pyruvate, which are linked to macrophage polarization in humans [132,133], are also important in zebrafish, where serine metabolism plays a crucial role in responding to bacterial infections [134]. Another enriched pathway, oxidative phosphorylation (Figure 8A), is implicated in macrophage polarization in humans [135], marking anti-inflammatory macrophages in contrast to pro-inflammatory macrophages, which rely on glycolysis. Additionally, during zebrafish tissue regeneration, macrophages initially exhibit glucocorticoid activation, followed by IL-10 signaling, and eventually, the induction of oxidative phosphorylation via IL-4/Polyamine signaling [118]. Our analysis further reveals that regulators of ferroptosis are specifically associated with zebrafish macrophages. While data on this process in fish immune cells are limited [136], in mammals, macrophages are crucial in regulating ferroptosis within tissues [137]. The enrichment of the term “Herpes simplex virus 1 infection” in macrophages is particularly noteworthy, as zebrafish are increasingly recognized as a valuable model for studying herpesvirus infections in mammals [138]. In humans, macrophages play a central role in the immune response against this virus [139], utilizing the enzyme acid ceramidase, which is localized in vesicles [140]. Recent studies indicate that acid ceramidase is also present in zebrafish and localizes to lysosomes [141], further supporting zebrafish as an effective model for investigating the antiviral activities of the vertebrate innate immune system.

We also identified the enrichment of common pathways in zebrafish macrophages and neutrophils (Figure 8B). In particular, the NOD-like receptor signaling pathway, which plays a crucial role in the antimicrobial activity of these cells across a wide range of multicellular organisms [142], was prominently enriched. Additionally, components of the lysosome, phagosome, and apoptosis pathways, all essential for the function of these cell types [143,144,145], were also significantly represented. Both macrophages and neutrophils exhibited a marked increase in the expression of components associated with heightened metabolic and energy activity, including carbon metabolism, metabolic pathways, and glycolysis/gluconeogenesis. For example, mammalian neutrophils store glycogen, which is critical for their survival and function, with the glycogen cycle being fundamental to the metabolism of these cells [146]. Collectively, these findings confirm that the innate immune cells analyzed in this study exhibit consistent transcriptomic signatures.

We observed specific enrichment of terms that reflect the unique biology of neutrophils (Figure 8C). Notably, genes involved in vascular smooth muscle contraction were found to be specific to this cell type. Both neutrophils [147] and macrophages are known to participate in the regulation of vascular smooth muscle cells in humans, with several molecular, genetic, and biochemical mechanisms underlying their adhesive interactions with these cells [148]. Neutrophils influence smooth muscle cells through the release of extracellular traps and are considered crucial players in the development of pathological conditions such as hypertension [147]. Additionally, monocytes, macrophages, and vascular smooth muscle cells are key contributors to the progression of atherosclerosis [149]. Our findings, along with existing data on the role of innate immune cells in the regulation of vascular smooth muscle cell proliferation [147] and cardiomyocyte proliferation during zebrafish cardiac regeneration [150], suggest that the interaction of immune cells with vascular smooth muscle cells in fish may occur similarly to that in humans, making zebrafish a promising model for studying a wide range of cardiovascular pathologies. Furthermore, the enrichment of neutrophil marker genes with components of the GnRH (gonadotropin-releasing hormone) signaling pathway in fish reveals parallels with the functional characteristics of these cells in humans. In humans, GnRH influences the formation of neutrophil extracellular traps and plays a role in processes such as wound healing in diabetes [151]. Although the effect of GnRH on neutrophils in fish has yet to be studied, these parallels suggest a similar functional presence. Additionally, among the pathways enriched in fish neutrophils, the three genes *odc1*, *pgd*, and *gsr* are responsible for glutathione metabolism. In mouse neutrophils, reduced glutathione is essential for maintaining their oxidative burst capabilities and antimicrobial activity [152]. The C-type lectin receptor signaling pathway is uniquely enriched in zebrafish neutrophils. In mice, activated dermal neutrophils, especially in cases of epidermolysis bullosa acquisita, show upregulation of multiple C-type lectin receptors [153]. Lastly, genes associated with adherens junctions and tight junctions were also significantly enriched in fish neutrophils. In mammalian neutrophils, these junctions play a critical role during inflammation and in neutrophil–epithelial interactions [154].

#### 3.3.4. Identified Molecular Profiles of Pro- and Anti-Inflammatory Immune Cells

In our dataset, the majority of macrophages and neutrophils are characterized by the expression of typical marker genes associated with mature immune cells of these types. A crucial functional aspect of these cells’ homeostasis is the lability of their transcriptional programs and their capacity to polarize into subtypes with contrasting properties. A simplified classification divides macrophages into pro-inflammatory M1 macrophages, which stimulate pathogen defense, and anti-inflammatory M2 macrophages, which facilitate wound healing and regeneration. However, M1-like human macrophages are involved in both tumor-promoting and anti-tumor processes, and recent studies have revealed extensive immunophenotypes and plasticity in both M1-like and M2-like cells [155]. A similar classification exists for neutrophils, dividing them into pro- and anti-inflammatory subtypes, with the main cytokines regulating their polarization being well-known [156]. In zebrafish, key genes expressed in M1 macrophages include *tnfa*, *tnfb*, *il1b*, and *il6*, while M2 macrophages are known to express *tgfb1*, *ccr2*, and *cxcr4b* at high levels [157]. Similarly, pro-inflammatory neutrophils in fish express *tnfa*, *tnfb*, *il1b*, and *il6* [158], whereas anti-inflammatory neutrophils are characterized by the expression of *sgk1* and *hif1a* [159].

Therefore, we sought to identify macrophage and neutrophil cells within our dataset that were strongly polarized toward either the pro-inflammatory or anti-inflammatory phenotype, aiming to reveal the typical molecular signatures of these distinct subpopulations. The lists of genes that differentiate these subpopulations from the broader immune cell population are provided in Appendix A. Among the entire macrophage community (15,365 cells), we identified 39 cells with an extreme pro-inflammatory phenotype, characterized by the high-level expression of all four markers (*il1b*, *il6*, *tnfa*, and *tnfb*) (marker genes for these cells are listed in Appendix A). However, we did not detect any macrophages with the M2 phenotype described previously, which would be indicated by the simultaneous non-zero expression of *tgfb1a*, *tgfb1b*, *cxcr4b*, and *ccr2*. Among neutrophils (26,556 cells), we identified a small subset of pro-inflammatory cells (27 in total) expressing *il1b*, *il6*, *tnfa*, and *tnfb*, with their marker genes provided in Appendix A. We also found a group of anti-inflammatory neutrophils (28 cells in total) that expressed *sgk1* and *hif1a* at the highest levels (Appendix A). These findings suggest that only a very small percentage of immune cells in our sample can be definitively categorized into one polarization phenotype or the other. The vast majority of mature immune cells in the fish kidney retain transcriptomic lability, supporting the notion that the binary classification of the innate immune system (IIS) into pro-inflammatory and anti-inflammatory cells may be outdated. This concept is further supported by recent studies highlighting the overall immunophenotypic plasticity of these cells [155,160]. Although this classification of immune cells has its limitations, we further examined which of the macrophage-specific (Section 3.3.1) and neutrophil-specific (Section 3.3.2) markers identified in our study are associated with pro- or anti-inflammatory processes.

Macrophage-specific genes: Ten identified markers were significantly upregulated in M1 macrophages, including *ctsz*, *vmp1*, *tspan36*, *sftpbb*, *ccl34a.4*, *lgmn*, *fgl2a*, *mfap4.1*, *sdc4*, and *lgals3bp.1*. Additionally, three potential macrophage markers, *atf4a*, *pglyrp5*, and *fthl27*, were upregulated across all three polarized immune populations (M1 macrophages, pro-inflammatory neutrophils, and anti-inflammatory neutrophils), suggesting their critical role in the zebrafish innate immune response. The well-known macrophage-specific gene *lygl1* and transcription factor *junba* were not upregulated during M1 macrophage polarization but were instead expressed in both pro-inflammatory and anti-inflammatory neutrophils. Furthermore, *nr4a1*, *ckbb*, and *ctsd* were specifically upregulated in pro-inflammatory neutrophils, while the transcription factor *nfe2l2a* was upregulated in anti-inflammatory neutrophils.

Neutrophil-specific genes: Six potential neutrophil marker genes were significantly upregulated in all polarized subsets, including *il1b*, *rgs2*, *atf3*, *hif1ab*, *stat1a*, and *ncf1*. This set intriguingly includes both the pro-inflammatory marker *il1b* and the anti-inflammatory *hif1ab*, which suggests that transcriptional regulation of immune cell polarization in zebrafish may not be strictly discrete but instead may involve a gradient expression of key interleukins and mediators. Such insights are crucial for constructing more accurate cell-based models in the future, particularly when relying on scRNA-seq data [161]. The transcription factors *atf4b* and *lrrfip1a*, as well as the well-known marker *mpx*, were upregulated in both polarized subsets of neutrophils. Additionally, the gene *srgn* and transcription factor *id3* were upregulated in M1 macrophages. The transcription factor *plek* was upregulated in both anti-inflammatory neutrophils and M1 macrophages, while the genes *amd1* and *hsd3b7* were specifically upregulated in anti-inflammatory neutrophils.

In summary, a small proportion of the markers identified in our gene networks are specifically associated with cell polarization. Additionally, several marker genes show upregulated expression during the polarization of another cell type, suggesting their functional significance in both neutrophils and macrophages. It is remarkable that the majority of macrophage (39 out of 62) and neutrophil (25 out of 43) markers did not exhibit differential expression during polarization, indicating their universal roles in mature immune cells. The high expression of these genes appears to be less dependent on the polarization state of the cell. Therefore, we propose that future studies should shift their focus from the individual properties of specific immune cell populations and subtypes to the complex interactions and regulatory mechanisms governing entire communities and subtypes of fish immune cells across various processes. As highlighted in a recent review, there is a notable gap in research on the cooperation and interactions among zebrafish neutrophils and macrophages [6]. Our meta-analysis underscores the dominant role these immune cells play in the zebrafish IIS and reveals that their classification is far more complex than the simple dichotomy of pro- and anti-inflammatory cells.

## 4. Conclusions

Our study demonstrates the effectiveness of meta-analysis as an approach to identifying key properties of large heterogeneous immune cell populations. The flexibility and iterativeness of this strategy in filtering cell types, identifying cell types and marker genes, and the ability to combine such data with interactomic analysis make it a powerful tool for predicting molecular–genetic systems of certain cell populations. In total, we identified more than two hundred highly expressed genes for macrophages and neutrophils of zebrafish kidneys. For more than half of the newly identified marker genes of them, we found literature evidence of the function of its direct or indirect ortholog in the same immune types in humans or mice. These data are still fragmentary and allow only a rough statement about the general similarity of mammalian and fish immune cells, but they can become the basis for further detailed immunogenetics and evolutionary studies. Also, macrophages and neutrophils account for about half of the cells in the immune dataset, highlighting their prevalence and importance for fish immunity. Our analysis also highlights the high complexity and heterogeneity within these cell types and the weakness of their binary classification into pro- and anti-inflammatory phenotypes.

On the other hand, meta-analysis is not an optimal tool for studying rare cell types. Small cell populations may be excluded or mistakenly merged with other cell subtypes during such integrative analyses, making their identification and annotation difficult. Our analysis did not detect zebrafish eosinophils and NK cells. To study the functional roles and features of such minor cell types, separate special experiments are needed, including additional steps such as cell sorting, immunostaining, and probably the usage of specific transgenic fish lines.

One of the main limitations of scRNAseq data is low read coverage. Our analysis involved about 250,000 immune cells, less than half of which passed the medium-stringency filtering threshold. Therefore, the integrative analysis of large scRNAseq datasets cannot yet guarantee accurate prediction of more than several hundred marker genes per cell type. Our results reflect only the highest expressed genes of these cell types. In turn, markers with low counts per cell cannot be considered unambiguous, and they require additional validation. We could try to bypass this limitation by merging individual cells, considering them metacells. However, this approach is quite slippery since it introduces significant Bayes associated with the specific characteristics of the analyzed dataset and does not guarantee that low-expressed marker genes identified by this method can be identified in another experiment. However, single-cell technologies continue to rapidly develop, and this issue could be overcome in the next few years.

## Figures and Tables

**Figure 1 biology-13-00773-f001:**
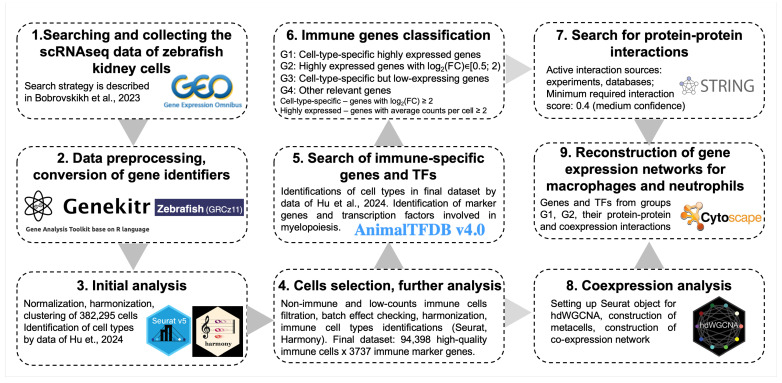
Key steps of the conducted scRNA-seq meta-analysis. The first stage of the analysis consisted of searching and selecting data from 13 experiments from the GEO NCBI database in the form of count matrices or Seurat objects. The second stage consisted of bringing the data into a uniform format, combining individual subsets of experiments into single matrices, and converting gene identifiers into a GRCz11 format using Genekitr [22]. Next, the procedures for integrating the 13 matrices, normalizing them, harmonizing them to eliminate the batch effect, and clustering were carried out using Seurat v. 5 [23] and Harmony [24]. The fourth stage consisted of filtering out non-immune cells, cells with low coverage, and genes irrelevant to immune cells. After filtering, the harmonization and clustering procedures were repeated on the final dataset. The fifth stage was to determine the cell types of immune cells in the final dataset using the main known marker genes from the recent work [8]. Transcription factors were identified according to AnimalTFDB4 [25]. Next, in stage six, the marker genes of the identified cell types were calculated and classified by expression level and fold changes in target types. Stages 7–9 consisted of reconstructing gene networks for macrophage- and neutrophil-specific genes using the STRING-db v. 12.0 database and co-expression data calculated from our final dataset using the hdWGCNA [26] package. Network layout and visualization were performed using Cytoscape v. 3.10.2 [27].

**Figure 2 biology-13-00773-f002:**
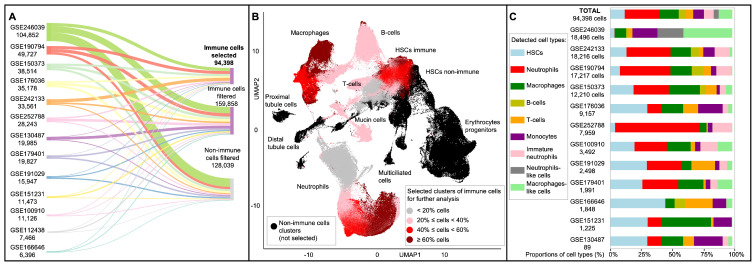
Primary steps of zebrafish kidney marrow scRNA-seq dataset integration and IIS cell classification. (**A**) Sankey diagram showing the total number of zebrafish kidney cells pooled from each of the 13 selected experiments. The initial analysis included 382,295 cells that passed the soft filtering threshold (nFeatureRNA greater than 200 and the percent of reads mapped to mitochondrial genes less than 10%) and were further classified into immune (a total of 254,256 cells) and non-immune (a total of 128,039 cells) categories. The final analysis included 94,398 immune cells from the selected group, characterized by high quality and satisfying the following criteria: 3000 ≤ nCountRNA ≤ 30,000 and 400 ≤ nFeatureRNA ≤ 4000. (**B**) Distribution of the selected 94,398 cells within the original dataset of 382,295 cells. The color indicates cluster characteristics after filtration. Cells from clusters marked in black were excluded from further analysis as they were not immune, corresponding to the bottom stream in the Sankey diagram (**A**). Clusters with less than 20 percent of cells remaining after filtration are marked in gray. Clusters with 20 to 40 percent of cells remaining are marked in pink. Clusters with 40 to 60 percent of cells remaining are marked in red. Clusters with more than 60 percent of cells remaining are marked in dark red. Cell types were labeled according to the key markers from [8]. Expression plots of the main marker genes are provided in Appendix A. (**C**) Total number of cells selected from individual experiments and their cell-type composition. Stacked bar charts display the overall composition of the datasets by cell type. HSCs refer to hematopoietic stem cells.

**Figure 3 biology-13-00773-f003:**
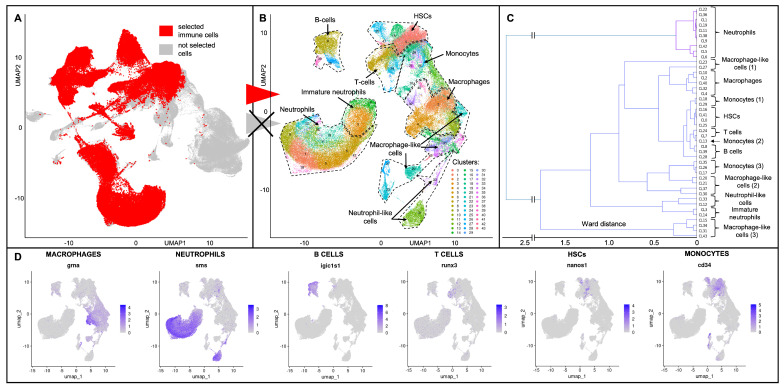
General information on the final dataset of the selected IIS cells. (**A**) Distribution of the selected immune cells, totaling 94,398, in the harmonized UMAP space containing 382,295 cells. The selected immune cells are marked in red within the original dataset space. (**B**) Distribution of selected zebrafish kidney immune cells in the harmonized UMAP space of immune-specific differentially expressed genes, comprising 94,398 cells. A total of 44 identified Seurat clusters are numbered in descending order based on their cell count. The areas corresponding to the identified cell types are outlined by a dashed line. HSCs refer to hematopoietic stem cells. (**C**) Hierarchical clustering dendrogram of immune cell clusters based on the expression of immune-specific differentially expressed genes. This clustering is based on the correlating gene expression of 3737 immune genes between Seurat clusters. The separated subgroups of the same cell types are numbered with (1)–(3). (**D**) Distribution of the expression of key marker genes, selected based on [8], used to annotate the main immune cell types.

**Figure 4 biology-13-00773-f004:**
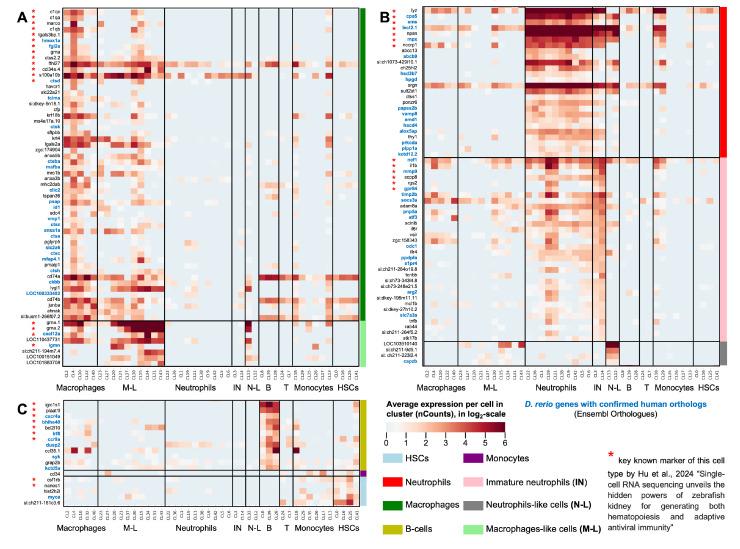
The expression heatmap for identified cell-type-specific highly expressed marker genes in different cell types. The heatmap displays highly expressed cell-type-specific genes with log2FC ≥ 2 and average counts-per-cell (ACPC) ≥ 2 in the target cell type. The expression values of genes in 44 clusters, grouped by cell types, are shown using a blue–red color scheme. A colored bar to the right of the heatmap indicates the cell type to which the gene is specific. The cell type labels are provided at the bottom of the heatmap. Red asterisks mark known marker genes for this cell type, as reported by [8]. Common names for *Danio rerio* genes are provided; genes with high-confidence human orthologs are marked in bold blue font. (**A**) Identified 62 marker genes for macrophages and macrophage-like cells. (**B**) Identified 62 marker genes for mature neutrophils, immature neutrophils, and neutrophil-like cells. (**C**) Identified 18 marker genes specific to other immune cell types (B cells, monocytes, and hematopoietic stem cells).

**Figure 5 biology-13-00773-f005:**
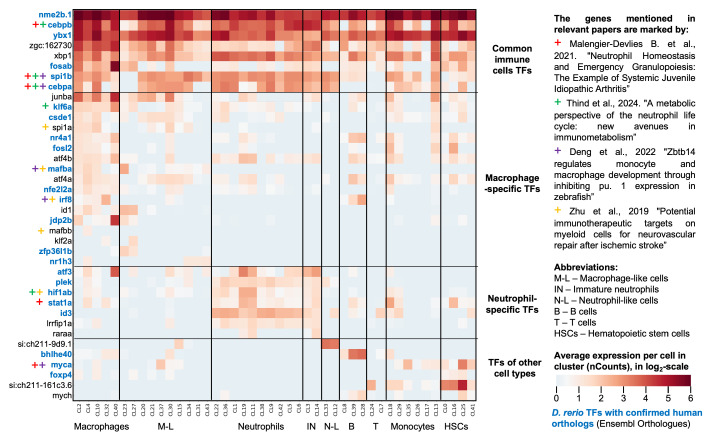
Heatmap of identified transcription factors (TFs) expressed in at least one immune cell cluster in zebrafish kidney (ACPC ≥1). Gene names are listed on the left, with those having the Homo sapiens ortholog(s) with high confidence (according to Ensembl Orthologues) highlighted in bold blue. Genes that were previously mentioned in related papers [105,106,107,108] are marked with plus signs with certain colors. The expression values of genes across 44 clusters, grouped by cell types, are depicted using a blue–red color scheme. TFs are categorized based on their expression patterns into sections from top to bottom: common (expressed in multiple identified immune cell types), macrophage-specific, neutrophil-specific, and TFs of other cell types. Cell type labels are provided at the bottom of the heatmaps.

**Figure 6 biology-13-00773-f006:**
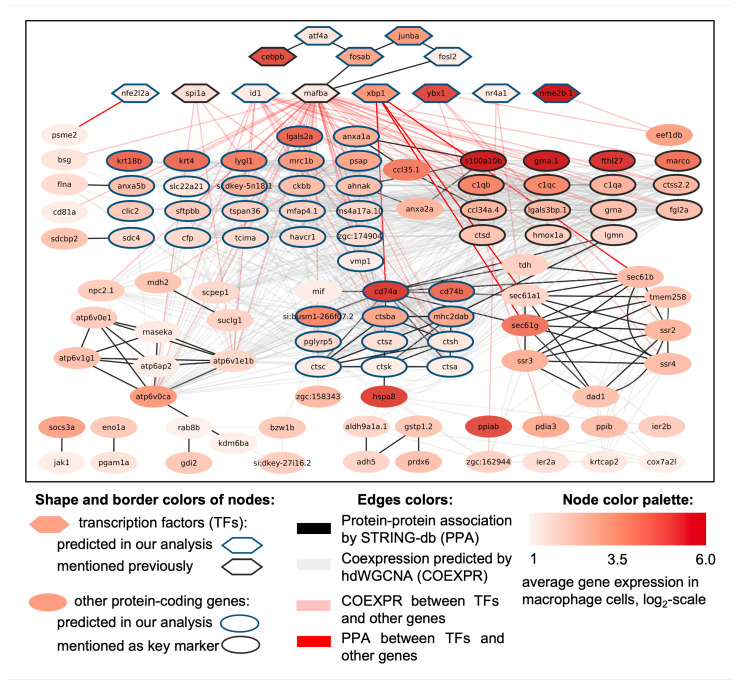
Reconstructed gene network of highly expressed genes in the population of zebrafish kidney macrophages. Nodes in the network represent either transcription factors (hexagons) or other protein-coding genes (ellipses). The red fill intensity of the nodes indicates the level of ACPC expression within the macrophage population. Ellipses are highlighted to denote highly specific macrophage genes (log2FC ≥ 2): those outlined in black represent key macrophage marker genes confirmed by [8], while those outlined in blue denote potential new macrophage markers. Transcription factors previously identified in the literature [105,106,107,108] are outlined in black, whereas those identified in our analysis are outlined in blue. The color of the connections represents the type of interaction: black lines indicate protein–protein associations based on STRING-db v. 12.0 (minimum required interaction score: 0.4 (medium); sources of active interactions: experiments and databases); gray lines denote co-expression interactions predicted by our analysis using hdWGCNA (minimum co-expression score = 0.1); red lines represent protein–protein associations between transcription factors and protein-coding genes; pink lines indicate co-expression among transcription factors and protein-coding genes.

**Figure 7 biology-13-00773-f007:**
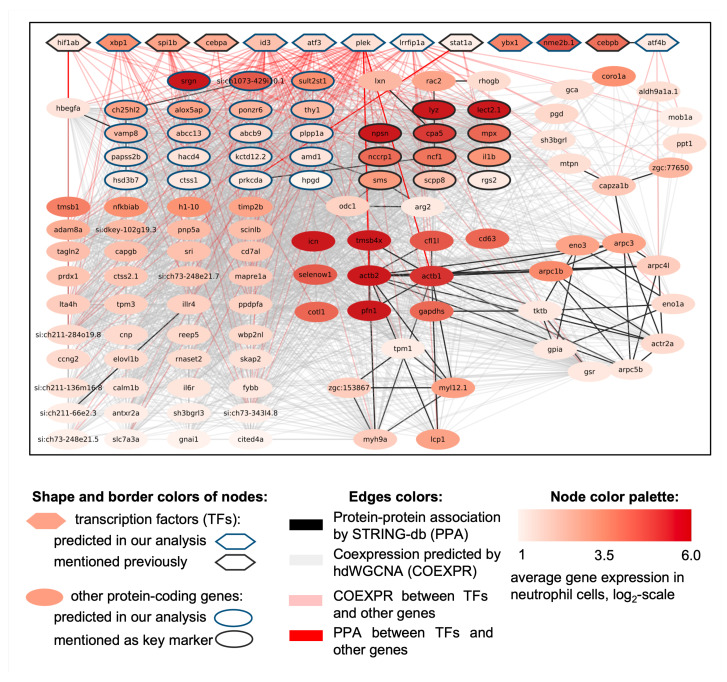
Reconstructed gene network of highly expressed genes in the population of zebrafish kidney neutrophils. Nodes in the network represent either transcription factors (hexagons) or other protein-coding genes (ellipses). The intensity of the red fill in the nodes reflects the level of ACPC expression within the neutrophil population. Ellipses are highlighted to denote highly specific neutrophil genes (log2FC ≥ 2): those outlined in black are key neutrophil marker genes confirmed by [8], while those outlined in blue represent potential new neutrophil markers. Transcription factors previously identified in the literature [105,106,107,108] are outlined in black, while those identified in our analysis are outlined in blue. The color of the connections indicates the type of interaction: black lines represent protein–protein associations based on STRING-db (minimum required interaction score: 0.4 (medium); sources of active interactions include experiments and databases); gray lines denote co-expression interactions predicted by our analysis using hdWGCNA (minimum co-expression score = 0.1); red lines indicate protein–protein associations between transcription factors and protein-coding genes; pink lines represent co-expression among transcription factors and protein-coding genes.

**Figure 8 biology-13-00773-f008:**
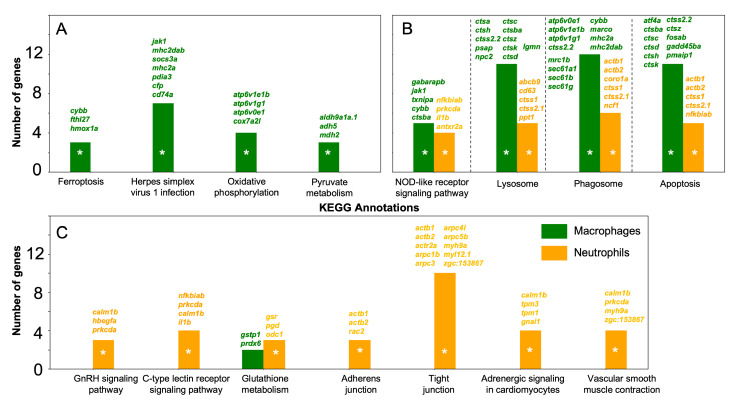
Specific KEGG pathways enriched in macrophages and neutrophils. Green bars and green genes represent macrophage-specific pathways, while orange bars and orange genes represent neutrophil-specific pathways. Significant enrichment within these pathways is indicated by white asterisks. The names of the genes included in each pathway are displayed above the corresponding bars. (**A**) Macrophage-specific enriched pathways; (**B**) commonly enriched pathways; (**C**) neutrophil-specific enriched pathways.

**Table 1 biology-13-00773-t001:** Single-cell transcriptomics experiments of zebrafish kidney used in the meta-analysis.

GEO NCBI ID ^1^	Age	Genotype/Strain	Number of Used Samples	Initial Number of Cells	Ref.
GSE100910	3–9 months	WT; *prkdc*	6	11,327	[29]
GSE112438	not available	AB; CD41:GFP	37	13,824	[30]
GSE130487	4–12 months	WT	1	20,000	[31]
GSE150373	8 months	WT; *runx1*	8	39,424	[32]
GSE151231	4 months	WT; *gata2b*	9	14,463	[33]
GSE166646	adult	WT	1	6422	[34]
GSE176036	8 months	*runx1*	8	35,178	[32]
GSE179401	2 months	WT; *rag*; *rag il2rga*	9	47,832	[35]
GSE190794	4 months	GESTALT	10	51,540	[36]
GSE191029	adult	WT; *prkcda*; *cxcl8*	8	20,695	[37]
GSE242133	1–1.5 months	AB	3	36,600	[8]
GSE246039	3 months	WT; *cebpb*	10	105,194	[38]
GSE252788	6 months	*cebpb*	2	28,534	[9]

^1^ Detailed information about these experiments is provided in Appendix A.

**Table 2 biology-13-00773-t002:** IIS cell types and their presence in Seurat clusters identified in our meta-analysis.

Cell Type	Total Cell Number	Seurat Cluster No	Cell Number in Cluster
HSCs	11,013	0	7852
		16	1736
		25	1202
		41	223
Macrophages	15,365	2	5831
		4	5232
		10	3149
		32	877
		40	276
Macrophage-like cells	10,595	15	1769
		20	1600
		21	1552
		23	1545
		27	992
		30	910
		31	879
		34	819
		37	408
		43	121
Neutrophils	26,556	1	6372
		5	4812
		6	4466
		9	3506
		11	3095
		19	1684
		22	1550
		36	486
		38	403
		42	182
Immature neutrophils	7477	3	5383
		14	2094
Neutrophil-like cells	3876	12	3035
		33	841
B cells	5416	8	4137
		28	982
		39	297
T cells	5678	7	4209
		24	1469
Monocytes	8422	13	2334
		17	1710
		18	1705
		26	1019
		29	960
		35	697

## Data Availability

Data used in this study are available from a publicly accessible repository Gene Expression Omnibus database. The identifiers of particular experiments and corresponding articles are available in Table 1. The joined expression matrix, which was used in our final analysis and consisted of 94,938 immune cells × 3737 immune-related genes, is provided in Appendix A. Other additional materials and results of our calculations can be provided to those interested upon reasonable request.

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
