# Peer review of "Catching the Big Fish in Big Data: A Meta-Analysis of Zebrafish Kidney scRNA-Seq Datasets Highlights Conserved Molecular Profiles of Macrophages and Neutrophils in Vertebrates"

_biology, 2024, doi:10.3390/biology13100773_

Round 1

Reviewer 1 Report

Comments and Suggestions for Authors

Title: Catching the big fish in big data: a meta-analysis of zebrafish kidney scRNA-seq datasets highlights conserved molecular profiles of macrophages and neutrophils in vertebrates

Journal: Biology-3181572

The authors well designed the overall manuscript, the methodology was appropriate, and collected all the information was correct and meaningful. Hopefully, this manuscript will have a great contribution to the scientific community. However, I have minor corrections in the manuscript to improve the quality. I will suggest to the authors to include some images of HSCs, macrophages, neutrophils, monocyte, T cell, B cell etc.  

Author Response

Dear reviewer,

We sincerely thank you for your high appreciation of our work and manuscript.

Thank you for your suggestion to include images of fish immune cells; this would be really interesting, and we are thinking about such works in the future. However, we do not have high-quality uniform images of zebrafish immune cells at the moment, and we wanted to focus on the genetic systems in our study. And with adding detailed morphology of immune cells, we are afraid that our already large article will become even larger and may confuse readers.

During revision, we significantly improved the overall quality of our paper. We will be happy to answer your questions if you have any.

Reviewer 2 Report

Comments and Suggestions for Authors

The paper Catching the big fish in big data: a meta-analysis of zebrafish kidney scRNA-seq datasets highlights conserved molecular profiles of macrophages and neutrophils in vertebrates   conducted a meta-analysis of single-cell RNA sequencing (scRNA-seq) datasets from zebrafish kidney marrow. Its an interesting study, some major point are stated below

1. Section 2.1 data collection, the data were collected from different experiments reported previously. However, the conditions in each experiments were different, and different genes may expressed under special conditions, how to eliminate the impact of these different.

2. How to define marker genes?

3. For classification of cell-type-specific genes, is it reliable? Or are there any references to support this?

4. Besides macrophages and neutrophils, the author also analyzed the Hematopoietic stem cells, B cells, T cells, and monocytes, so is the topic of this paper appropriate?

5. Table 2, what did the numbers in cluster No represent.

6. Some bold font is not necessary.

7. The reference number should be reduce appropriately

Author Response

Dear reviewer,

We sincerely thank you for your high appreciation of our work and manuscript.
We have taken your comments into account and significantly improved our paper.

Below we provided the answers (A) to your questions (Q).

Q1: Section 2.1 data collection, the data were collected from different experiments reported previously. However, the conditions in each experiments were different, and different genes may expressed under special conditions, how to eliminate the impact of these different.
A1: Yes, it was the key idea of our meta-analysis—to identify the most stable markers of various immune cells in zebrafish. We include cells from different ages and genotypes of zebrafish in experiments, and our integration results suggest that the main IIS cell types among them are pretty conservative. In addition, we effectively eliminate the impact of individual differences by using normalization and harmonization procedures during our bioinformatic analysis.

Q2. How to define “marker genes”?

A2: Marker genes of immune cells in our study are defined by functions and thresholds described in Section 2.2.3.
Immune-related marker genes were calculated using the function FindMarkers with the following parameters: only.pos = TRUE (upregulated), min.pct = 0.02 (expressed at least in 2 percent cells of the immune population), and logfc.threshold = 0.5 (upregulated by log2FC ≥ 0.5 in the immune population);
Markers of individual Seurat clusters of immune cells were calculated using the function FindAllMarkers with the following parameters: only.pos = TRUE (upregulated), min.pct = 0.25 (expressed at least in 25 percent cells of target clusters), and logfc.threshold = 0.5 (upregulated by log2FC ≥ 0.5 in target clusters compared to all other clusters);
Marker genes of cell types were calculated using the function FindMarkers with the following parameters: only.pos = TRUE (upregulated), min.pct = 0.25 (expressed at least in 25 percent cells of the target cell type), and logfc.threshold = 0.5 (upregulated by log2FC ≥ 0.5 in target cell type);
We have added these explanations in brackets to Section 2.2.3 for better clarity.

Q3. For “classification of cell-type-specific genes”, is it reliable? Or are there any references to support this?
A3: We renamed this section to not confuse the readers to "Classification of cell-type-specific markers", because this is a further classification of obtained cell-type specific markers (described in the last paragraph of Section 2.2.3). We took rather stringent thresholds for both log2FC ≥ 2 (at least four times upregulated in cell type) and basal expression level (ACPC ≥ 2) to consider marker genes as belonging to the first group (as cell-type-specific genes). It is this group that we further considered in our results as potential new cell type markers that are most prominently expressed in the target cell types. This classification is empirical and is supported by literature data. In particular, Group 1 of macrophage genes included 13 of the top 20 macrophage markers from the work (Hu et al., 2024), and we also confirmed that 13 of the top 20 markers of this cell type belong to Group 1 of neutrophils (Hu et al., 2024). In addition, we placed this explanation in Section 3.2. of our results:

"We identified 62 highly specific marker genes for macrophages and macrophage-like cells (see Figure 4A), thirteen of which were previously highlighted in the list of the 20 most specific macrophage marker genes \cite{hu2024single}. For mature neutrophils and immature neutrophils, 58 of such specific genes were revealed, depicted at Figure 4B; among them are thirteen most relevant neutrophil genes mentioned previously \cite{hu2024single}. B-cells and HSCs can be identified by their specific twelve and five markers, respectively, shown in Figure 4C. Seven out of twelve B-cell markers previously reported by Hu et al. as well as two markers of HSCs, which were validated in our analysis \cite{hu2024single}. Therefore, our empirical classification is reliable since we used rather stringent thresholds for both fold-changes and basal expression level for selecting these genes. Furthermore, our classification and selection of these genes is strongly supported by previous data \cite{hu2024single} and expands key marker genes for neutrophils and macrophages in the zebrafish kidney."

Q4. Besides macrophages and neutrophils, the author also analyzed the Hematopoietic stem cells, B cells, T cells, and monocytes, so is the topic of this paper appropriate?
A4: Yes, we analyzed all the immune cell types detected by us in the fish kidney, however, the greatest specificity and maturity, as well as the distinctiveness of the expression profiles, were demonstrated by neutrophils and macrophages, which is reflected in Figure 4 (cell-type-specific highly expressed marker genes). Therefore, we performed a detailed analysis and literature comparison specifically for these cell types. For other immune cell types, our analysis did not reveal so many stable marker genes to conduct a systematic comparison of their molecular genetic systems with known data on the mammalian immune system. Therefore, we consider the topic of our paper as appropriate.

Q5. Table 2, what did the numbers in “cluster No” represent.
A5: It is the index number of the Seurat cluster where this immune cell type is identified. Thank you; we have improved the description of this table.

Q6. Some bold font is not necessary.
A6: We checked the text and removed some unnecessary bold fonts.

Q7. The reference number should be reduce appropriately
A7: We appreciate your feedback and removed some non-important references. However, we believe that a detailed discussion of the genes and genetic systems we have discovered is critical to understanding the biology of fish immune cells. Such a discussion provides a detailed comparison of fish and mammalian immune systems that may be useful for further experimental design and the development of novel applications of zebrafish as a model of human immunity.

During revision, we significantly improved the overall quality of our paper. We will be happy to answer your questions if you have any.

Reviewer 3 Report

Comments and Suggestions for Authors

‘Catching the big fish in big data: a meta-analysis of zebrafish kidney scRNA-seq datasets highlights conserved molecular profiles of macrophages and neutrophils in vertebrates' by Bobrovskikh et al.

This manuscript presents the meta-analysis of single-cell RNA sequencing (scRNA-seq) datasets from zebrafish kidneys. Overall, the information presented is of value and provides detailed data to explore the innate immune system further. However, a few minor comments have to be addressed before acceptance for publication.

· A minor level of English language editing has to be done for this manuscript.

·       Line 34 – “…..with sophisticated fish leukocytes…..” – please rewrite/improve this part

·       Table 1 can display more information about the single-cell transcriptomics experiments such as the life stage of zebrafish used in the study eg. Larve or adult…; what type of experimental conditions the animals were subjected to…; and any other relevant information.

·       Line 183 – The authors have studied the immune-specific differential expression. However, it is not clear what conditions (contrasts) were compared. Please make that part more clear.

·       Line 868 – “…12 experiments…..” -  is it not 13 experiments???…..

Comments on the Quality of English Language

A minor level of English language editing has to be done for this manuscript.

Author Response

Dear reviewer,
We sincerely thank you for your high appreciation of our work and manuscript.
We did some minor English language editing according to your comments.

Below we provided the answers (A) to your questions (Q).

Q1: Line 34 – “…..with sophisticated fish leukocytes…..” – please rewrite/improve this part
A1: We rewrite this sentence as follows:
The overall antiquity and evolutionary conservatism of the IIS make zebrafish a potentially powerful model for studying animal pathogenesis, injuries, and wound healing; macrophages and neutrophils are known to play a crucial role in these processes.

Q2: Table 1 can display more information about the single-cell transcriptomics experiments such as the life stage of zebrafish used in the study eg. Larve or adult…; what type of experimental conditions the animals were subjected to…; and any other relevant information.
A2: Thank you! We added information about genotypes and life stages of zebrafish used in these experiments in Table 1.

Q3: Line 183 – The authors have studied the immune-specific differential expression. However, it is not clear what conditions (contrasts) were compared. Please make that part more clear.
A3: Thank you; we rewrote this part to be more clear for readers. We rephrased this part as follows:
"To precisely investigate the cell types of the revealed immune cell subset, we compared expression within all Seurat clusters to assess their similarity. In particular, the function AverageExpression was used on an immune cell subset to calculate the expression matrix of all Seurat clusters in average counts-per-cell (ACPC) values (matrix of Seurat clusters x ACPC values of immune-related marker genes). This matrix was used to calculate the correlation matrix using the pandas.DataFrame.corr function. Based on the correlation matrix, we calculated the distance matrix (1 - abs(correlation)). For hierarchical clustering, the square form of the distance matrix was given to the function scipy.cluster.hierarchy.linkage with the following parameters: method=’ward’, metric=’euclidean’, optimal ordering=’True’."

Q4: Line 868 – “…12 experiments…..” -  is it not 13 experiments???…..
A4: Thank you; this part is right. In particular, during the subsequent step of harmonization and clustering of 13 experiments, it was noted that the immune cells of the GSE112438 experiment revealed significant differences in embeddings compared to the cells from other experiments. This led to the exclusion of this experiment from the final analysis. This is written in Section 3.1. of the Results and Discussion.

During revision, we significantly improved the overall quality of our paper. We will be happy to answer your questions if you have any.

Reviewer 4 Report

Comments and Suggestions for Authors

Comments:

This study carried out an extensive meta-analysis of scRNA-seq datasets derived from zebrafish kidney marrow, focusing on the identification and classification of immune cells. It revealed that although the presence of key genetic pathways in zebrafish innate immune cells are similar to those identified in mammals, numerous genes that are specific to zebrafish neutrophil or macrophage lineages. This is a well-written paper containing interesting results which merit publication. For the benefit of the reader, however, following questions should be answered.

Specific comments:

Keywords

1. You should reduce the number of keywords to less than six to highlight the main idea.

Introduction

1. Lines 36-71: In this paragraph, various cell types contained in fish IIS are listed in detail, which is superfluous. Authors should keep only some important information to better serve the main idea.

2. Lines 110-122: In this paragraph, authors should emphasize the purpose and significance of the research rather than the results.

Materials and methods

1. Line 137: Its unclear who provided the count matrices.

2. Line 219: Groups 1 and Groups 2 should be Group 1 and Group 2.

3. Lines 244-251: The method description is repeated.

Results and Discussion

1. Lines 325 and 333: The abbreviation for hematopoietic stem cells is given on line 266, and only the abbreviation is needed here. Please check the full text for similar abbreviation problem.

2. Line 384: (2024) should be deleted.

3. Line 411: Which two are the major histocompatibility complex class II genes?

4. Line 431: Large yellow croaker should be given a Latin name. Please check the full text for missing Latin names of species.

5. Lines 493 and 504: Whether arg2 and ARG2 is the same gene? Please keep the writing consistent. And similar questions require full-text checking.

Conclusions

1. This section should succinctly and accurately summarize the important findings of this study, rather than boring the readers with a long text.

References

1. There are too many references, although most of them are recent studies. I think it is necessary to cut out some non-essential.

2. There are a few mistakes in the reference list, including page number errors, inconsistent of the capital letters, and problems with use of italics etc. Such as:

Line 923: Inconsistent use of capital letters in journal titles.

Line 954: Page number indicates incorrect.

Line 980, 990: Aeromonas veronii, Arabidopsis thaliana should be italicized.

Line 1009: Lack of page numbers.

Comments on the Quality of English Language

Comments:

This study carried out an extensive meta-analysis of scRNA-seq datasets derived from zebrafish kidney marrow, focusing on the identification and classification of immune cells. It revealed that although the presence of key genetic pathways in zebrafish innate immune cells are similar to those identified in mammals, numerous genes that are specific to zebrafish neutrophil or macrophage lineages. This is a well-written paper containing interesting results which merit publication. For the benefit of the reader, however, following questions should be answered.

Specific comments:

Keywords

1. You should reduce the number of keywords to less than six to highlight the main idea.

Introduction

1. Lines 36-71: In this paragraph, various cell types contained in fish IIS are listed in detail, which is superfluous. Authors should keep only some important information to better serve the main idea.

2. Lines 110-122: In this paragraph, authors should emphasize the purpose and significance of the research rather than the results.

Materials and methods

1. Line 137: Its unclear who provided the count matrices.

2. Line 219: Groups 1 and Groups 2 should be Group 1 and Group 2.

3. Lines 244-251: The method description is repeated.

Results and Discussion

1. Lines 325 and 333: The abbreviation for hematopoietic stem cells is given on line 266, and only the abbreviation is needed here. Please check the full text for similar abbreviation problem.

2. Line 384: (2024) should be deleted.

3. Line 411: Which two are the major histocompatibility complex class II genes?

4. Line 431: Large yellow croaker should be given a Latin name. Please check the full text for missing Latin names of species.

5. Lines 493 and 504: Whether arg2 and ARG2 is the same gene? Please keep the writing consistent. And similar questions require full-text checking.

Conclusions

1. This section should succinctly and accurately summarize the important findings of this study, rather than boring the readers with a long text.

References

1. There are too many references, although most of them are recent studies. I think it is necessary to cut out some non-essential.

2. There are a few mistakes in the reference list, including page number errors, inconsistent of the capital letters, and problems with use of italics etc. Such as:

Line 923: Inconsistent use of capital letters in journal titles.

Line 954: Page number indicates incorrect.

Line 980, 990: Aeromonas veronii, Arabidopsis thaliana should be italicized.

Line 1009: Lack of page numbers.

Author Response

Dear reviewer,

We sincerely thank you for your high appreciation of our work and manuscript.
We have taken your comments into account and significantly improved our paper.

Below we provided the answers (A) to your questions (Q).

Q1. You should reduce the number of keywords to less than six to highlight the main idea.
A1. We reduced them and left five major keywords: innate immune system; transcriptomic meta-analysis; single-cell RNA sequencing; gene network; transcription factors

Introduction
Q1. Lines 36-71: In this paragraph, various cell types contained in fish IIS are listed in detail, which is superfluous. Authors should keep only some important information to better serve the main idea.
A1. Thank you. We substantially decreased the superfluous details in this paragraph and reworked the overall introduction to better serve the main idea of our paper.

Q2. Lines 110-122: In this paragraph, authors should emphasize the purpose and significance of the research rather than the results.
A2. Thanks, we rewrote this paragraph and emphasized our purpose and significance of our research as follows:
"Our work aims to bridge the current gap in understanding the core molecular genetic systems of macrophages and neutrophils in zebrafish compared to mammals. In this study, we identified novel marker genes for macrophages, neutrophils, and other cells of the IIS in zebrafish and identified transcription factors involved in myelopoiesis in fish. The genes and pathways we discovered require further study in the field of immunogenetics and transgenesis in zebrafish to evaluate the functions of the newly discovered transcription factors and marker genes for neutrophils and macrophages in pathogenesis, regeneration, and aging. Thus, our research contributes to the development of novel applications of zebrafish as a model of human immunity. In addition, our work methodologically complements existing strategies for meta-analysis of scRNAseq data. Our chosen dataset analysis strategy of initially soft filtering the data followed by more stringent filtering of cells used in the final analysis proved to be flexible and effective, providing a diversity of cell types and high read coverage. We believe that in future meta-analyses, researchers should select the threshold for filtering cells by the number of reads based on the quality of the original data rather than blindly using certain thresholds in data analysis pipelines. Therefore, the general logic of our meta-analysis can be used in future studies devoted to joined analysis of scRNAseq data."

Materials and methods

Q1. Line 137: It’s unclear who provided the count matrices.
A1. We rewrote this sentence: "For the further meta-analysis, we downloaded and used the count matrices or Seurat objects provided by the authors in the Supplementary Materials in the corresponding GEO NCBI records (available by their GEO NCBI identifiers)."

Q2. Line 219: “Groups 1 and Groups 2” should be “Group 1 and Group 2”.
A2. We corrected it.

Q3. Lines 244-251: The method description is repeated.
A3. We removed the repeated sentence.

Results and Discussion

Q1. Lines 325 and 333: The abbreviation for hematopoietic stem cells is given on line 266, and only the abbreviation is needed here. Please check the full text for similar abbreviation problem.
A1. Thank you; we removed redundant abbreviations and checked the full manuscript for this problem.

Q2. Line 384: (2024) should be deleted.
A2. Done.

Q3. Line 411: Which two are the major histocompatibility complex class II genes?
A3. We corrected it—genes are mhc2a, mhc2dab.

Q4. Line 431: Large yellow croaker should be given a Latin name. Please check the full text for missing Latin names of species.
A4. We replaced it with the Latin name Larimichthys crocea. We also checked the full text for such possible mistakes and corrected them.

Q5. Lines 493 and 504: Whether arg2 and ARG2 is the same gene? Please keep the writing consistent. And similar questions require full-text checking.
A5. Thank you. Throughout Sections 3.2.1. - 3.2.4., we used the accepted gene nomenclature spelling: human genes are marked by upper case, mouse genes are marked by title case, and zebrafish genes are marked by lower case. We have added an explanation of this in Section 3.2. We checked and corrected our spelling of genes throughout this part and full text.

Conclusions

Q1. This section should succinctly and accurately summarize the important findings of this study, rather than boring the readers with a long text.
A1. Thanks, we completely rewrote the conclusion sections.

References

Q1. There are too many references, although most of them are recent studies. I think it is necessary to cut out some non-essential.
A1. Thank you; we cut out some of them.

Q2. There are a few mistakes in the reference list, including page number errors, inconsistent of the capital letters, and problems with use of italics etc. Such as:

Line 923: Inconsistent use of capital letters in journal titles.

Line 954: Page number indicates incorrect.

Line 980, 990: Aeromonas veronii, Arabidopsis thaliana should be italicized.

Line 1009: Lack of page numbers.

A2. Thank you; we corrected this and other mistakes in our references.

During revision, we significantly improved the overall quality of our paper. We will be happy to answer your questions if you have any.